# Recognition of commensal bacterial peptidoglycans defines *Drosophila* gut homeostasis and lifespan

Taro Onuma[1,2☯], Toshitaka Yamauchi[1☯], Hina Kosakamoto[2], Hibiki Kadoguchi[3], Takayuki Kuraishi[3], Takumi Murakami[4], Hiroshi Mori[4], Masayuki Miura[1], Fumiaki Obata[1,2,5]*

1 Department of Genetics, Graduate School of Pharmaceutical Sciences, The University of Tokyo, Tokyo, Japan, 2 RIKEN Center for Biosystems Dynamics Research, Hyogo, Japan, 3 Faculty of Pharmacy, Institute of Medical, Pharmaceutical and Health Sciences, Kanazawa University, Kanazawa, Japan, 4 Department of Informatics, National Institute of Genetics, Shizuoka, Japan, 5 Laboratory of Molecular Cell Biology and Development, Graduate School of Biostudies, Kyoto University, Kyoto, Japan

☯ These authors contributed equally to this work.
* fumiaki.obata@riken.jp

**Data Availability Statement:** The RNAseq data has been deposited in DDBJ under the accession number DRA015054.

## Abstract

Commensal microbes in animals have a profound impact on tissue homeostasis, stress resistance, and ageing. We previously showed in *Drosophila melanogaster* that *Acetobacter persici* is a member of the gut microbiota that promotes ageing and shortens fly lifespan. However, the molecular mechanism by which this specific bacterial species changes lifespan and physiology remains unclear. The difficulty in studying longevity using gnotobiotic flies is the high risk of contamination during ageing. To overcome this technical challenge, we used a bacteria-conditioned diet enriched with bacterial products and cell wall components. Here, we demonstrate that an *A. persici*-conditioned diet shortens lifespan and increases intestinal stem cell (ISC) proliferation. Feeding adult flies a diet conditioned with *A. persici*, but not with *Lactiplantibacillus plantarum*, can decrease lifespan but increase resistance to paraquat or oral infection of *Pseudomonas entomophila*, indicating that the bacterium alters the trade-off between lifespan and host defence. A transcriptomic analysis using fly intestine revealed that *A. persici* preferably induces antimicrobial peptides (AMPs), while *L. plantarum* upregulates amidase peptidoglycan recognition proteins (PGRPs). The specific induction of these Imd target genes by peptidoglycans from two bacterial species is due to the stimulation of the receptor PGRP-LC in the anterior midgut for AMPs or PGRP-LE from the posterior midgut for amidase PGRPs. Heat-killed *A. persici* also shortens lifespan and increases ISC proliferation via PGRP-LC, but it is not sufficient to alter the stress resistance. Our study emphasizes the significance of peptidoglycan specificity in determining the gut bacterial impact on healthspan. It also unveils the postbiotic effect of specific gut bacterial species, which turns flies into a "live fast, die young" lifestyle.

**Funding:** This work was supported by AMED-PRIME to F.O. (JP17gm6010010 and JP20gm6310011), and partly by AMED-Project for Elucidating and Controlling Mechanisms of Aging and Longevity to M.M. (JP21gm5010001). This work was also partially supported by grants from the Japan Society for the Promotion of Science to T.K. (22H02570), and to F.O. (20H05726 and 22H02769), and grants from Japan Science and Technology Agency (JST)-FOREST program to T.K. (JPMJFR2063) and by YakultBio-ScienceFoundation to F. O. The funders had no role in study design, data collection and analysis, decision to publish, or preparation of the manuscript.

**Competing interests:** The authors declare no competing interests.

## Author summary

Microbiota plays a vital role in our health, but it can also have a negative impact on the lifespan of certain model organisms, such as the fruit fly Drosophila melanogaster. Despite its impact, the molecular mechanism behind how gut bacteria limits host lifespan remains unclear. In this study, we investigated the mechanism that one specific species of microbiota shortens lifespan and disrupts gut homeostasis of the aged flies, using a "fermented" fly diet. We found that the specific effects of this bacterium on fly healthspan were due to its capability of stimulating a specific receptor in the gut that recognizes peptidoglycan, a component of bacterial cell wall. Paradoxically, the same bacterium also increases stress resistance and defence against oral infection of a pathogen. Our study provides insight into the mechanisms underlying how certain members of the microbiota can lead to a "life fast, die young" lifestyle.

## Introduction

We live in symbiosis with many microorganisms. The gut microbiome plays an important role in physiology, metabolism, growth, behaviour, and the stress response [1–3]. However, it does not only benefit the host. It has been reported that germ-free animals, including nematodes, flies, and mice, live longer than conventional animals [4–6], especially when nutrients are abundant [7]. These studies indicate that the gut microbiota can be detrimental to lifespan. The molecular effectors of the gut microbiota on the host healthspan constitute a wide range of metabolites, proteins, and cell wall components [8]. However, it is not fully understood how each bacterial species in the gut impacts lifespan at the molecular level.

*Drosophila melanogaster* is a powerful model for studying the lifespan-microbiota relationship because of its relatively short lifespan and simple gut microbiome [9]. The bacterial community in *Drosophila* predominantly consists of *Lactobacillaceae* and *Acetobaceraceae*, including the genera *Lactiplantibacillus* and *Acetobacter* [10,11]. Several studies have identified bacteria-derived metabolites that limit lifespan. For example, uracil from *Gluconobacter morbifer* or *Levilactobacillus brevis* and lactate from *Lactiplantibacillus plantarum* are reported to shorten lifespan through reactive oxygen species [12,13]. The gut microbiota also affects two major lifespan-determining mechanisms, the insulin/IGF-1 signalling pathway and the mechanistic target of rapamycin pathway [14,15], although to what extent these pathways contribute to lifespan determination has not been proven.

In addition, one of the key mechanisms by which the gut bacteria shorten the host lifespan is the immune deficiency (Imd) pathway, which is homologous to the mammalian tumour necrosis factor (TNF) signalling pathway. The Imd pathway has two pattern recognition receptors for diaminopimelic acid (DAP)-type peptidoglycans (PGNs), PGRP-LC and PGRP-LE [16]. These receptors transmit signals to common downstream factors and eventually activate the transcription factor Relish [17]. Age-dependent activation of the Imd pathway is attributed to the gut microbiota, which causes age-related dysfunction in various tissues, such as the gut, Malpighian tubules, and brain [5,18,19]. Removing the gut microbiota or suppressing the Imd pathway can prevent age-related intestinal dysregulation, which leads to the extension of lifespan [20]. The majority of age-related transcriptomic alterations are reported to be attenuated in germ-free flies [21].

The magnitude of the immunostimulatory capacity reasonably varies by bacterial species. We have shown that at least three species of *Acetobacteraceae* strongly activate the Imd pathway and shorten the fly lifespan, while *L. plantarum* has weaker potency to do so [18,22,23]. However, the detailed mechanism by which bacterial factors result in differential levels of Imd

activation and hence alter the host lifespan remains elusive. In this study, we described how each gut bacterial species influences the fly physiology, transcriptome, and ageing by using bacteria-conditioned diets (BacDs). We found that bacterial products of *A. persici* shorten lifespan and increase the resistance to oxidant paraquat and oral infection with the pathogen *Pseudomonas entomophila*. Contrary to our initial assumption, the predominant mechanism of bacteria-specific effects on ageing and lifespan is not dependent on bacteria-derived metabolites but rather due to sensing the cell wall component PGN by the receptor PGRP-LC. Our data also suggested that *A. persici* PGN increases host defence against the pathogen but is not sufficient for enhancing resistance to paraquat. This study demonstrates how a gut bacterium shifts the trade-offs between the host defence capacity and lifespan.

## Results

### Preparation of a bacteria-conditioned diet

We previously isolated *A. persici* Ai and *L. plantarum* Lsi as two major bacterial species from the gut of a laboratory strain [23]. We found that *A. persici* Ai has stronger potency of Imd activation than *L. plantarum* Lsi, which might be one of the causes of the lifespan-shortening effect of *Acetobacteraceae* [18]. To further investigate the functional disparity between the two species, we needed to utilize gnotobiotic flies in which the associated microbiome is defined (e.g., monoassociation). Unlike in mice, gnotobiotic experiments in flies are not usually implemented in a "germ-free isolator". Therefore, one of the difficulties in studying ageing and measuring lifespan using gnotobiotic flies is the higher probability of contamination since their maintenance requires frequently flipping the flies into new vials throughout their lifespan.

To make this challenging experiment more technically accessible, we used a bacteria-conditioned diet (BacD), in which bacterial products, as well as bacteria *per se*, are abundantly present [18]. To prepare the diet, a standard fly diet was inoculated with bacterial isolates (or only the culture medium as a negative control). Incubation of the diet at 25°C for 24 hours allowed the bacterial species to proliferate approximately one hundred-fold (Fig 1A and 1B), reaching $19.1 \times 10^4$ (*L. plantarum*) or $5.14 \times 10^4$ (*A. persici*) CFU/mg food that is comparable to the typical bacterial numbers in the standard diet with wild type flies (~$10^{4-5}$ CFU/mg food in case of 1 week old Canton-S male flies) in our laboratory. In addition to the major species, we also investigated two minor bacterial species: *Gluconobacter* sp. Gdi, a strain of *Acetobacteraceae* isolated previously from flies with Imd activation [23], and a newly isolated *Lactobacillaceae*, *Leuconostoc* sp. Leui (see Methods).

First, we performed LC–MS/MS analysis to determine how this bacterial conditioning procedure changes the metabolites (i.e., nutrients) levels in the diet. As we previously reported, *L. plantarum* Lsi-conditioned diets reduced dietary purine levels (S1A Fig, [18]). This is also the case for *Leuconostoc* sp. Leui. As expected, these two lactic acid bacteria produced lactate (S1A Fig). In contrast, *A. persici* Ai and *Gluconobacter* sp. Gdi commonly produced the polyamine spermidine (S1A Fig). These metabolite alterations suggested that BacD can be utilised for understanding how bacteria and bacterial products influence their hosts. Before feeding the diets to the flies, an antibiotics cocktail was added to sterilize the conditioned diet, which prevents further proliferation of the bacteria and stops them from fermenting the diet (Fig 1A). The same antibiotic cocktail was added for the negative control to compare the effect of the bacterial components.

### *Acetobacter persici* Ai-conditioned diet shortened lifespan

Experiments using the conditioned diet were conducted using the following scheme. Adult flies were reared with a standard diet for two days post eclosion and then fed with antibiotic

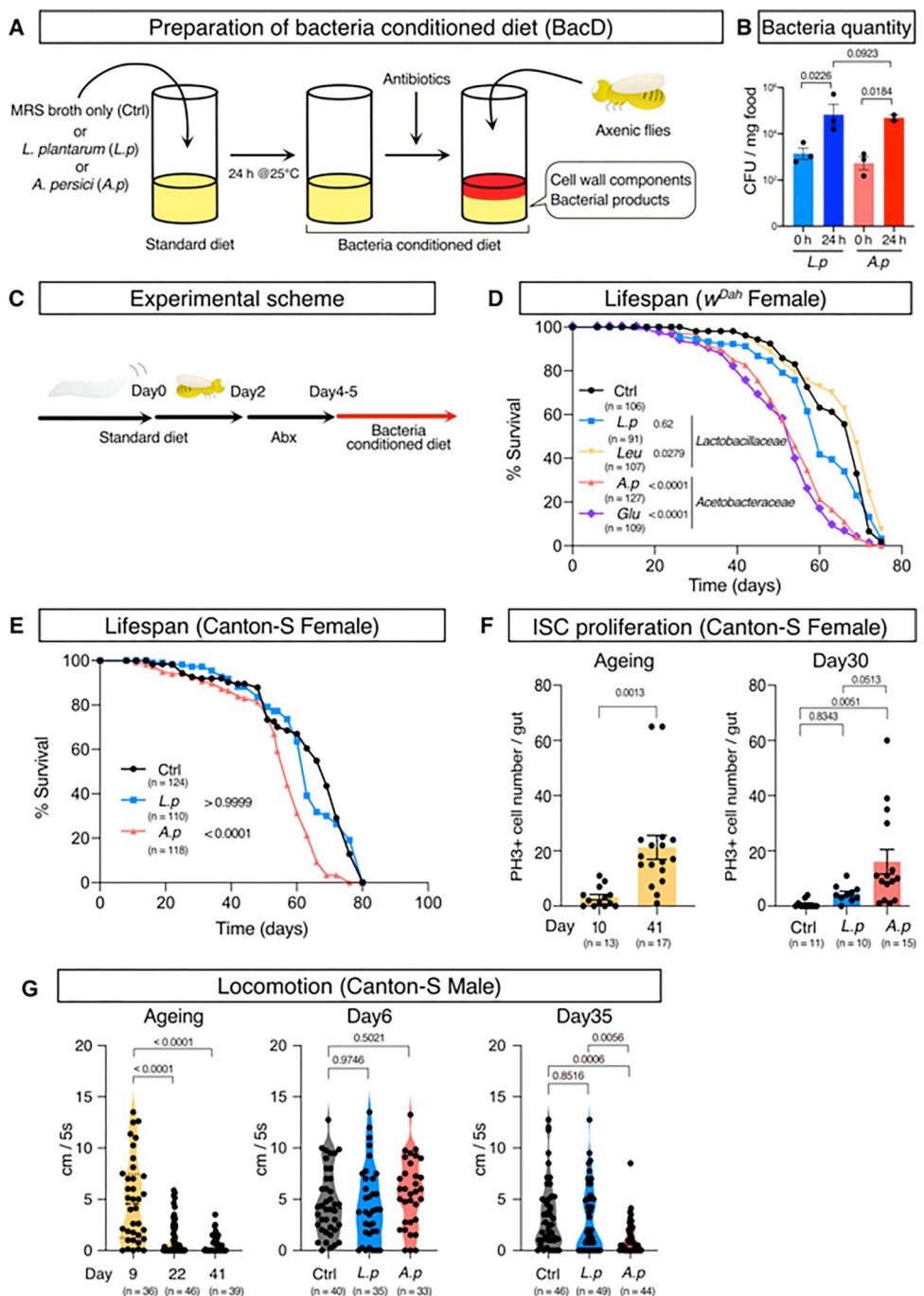

**Fig 1. *A. persici*-conditioned diet shortens lifespan.** (A) An overview of preparing the "bacteria conditioned diet (BacD)". (B) Colony forming units (CFU) of BacD (before addition of antibiotics). For the statistics, one-way ANOVA with Holm-Šídák's multiple comparison was used. (C) The experimental scheme of the ageing experiments using BacD. (D)(E) Lifespan of female $w^{Dah}$ (D) and Canton-S (E) flies with the BacD. A log-rank test was used to compare between control (Ctrl) and each BacD. (F) Phospho-histone H3-positive cell numbers in the midgut of Canton-S female flies fed a conventional diet (left, Day 10 vs Day 41) or 25 days of BacD (right, Day 30). For the statistics, unpaired two-tailed Student's *t* test (left figure) or one-way ANOVA with Holm-Šídák's multiple comparisons (right figure) was used. (G) The climbing ability of Canton-S male flies with either a conventional diet (Left, Day 9 vs Day 22 or Day 41) or BacD (Center, Day 6 or Right, Day 35) assessed by the negative geotaxis assay. For the statistics, unpaired two-tailed Student's *t* test (left figure) or one-way ANOVA with Holm-Šídák's multiple comparison (right figures) was used. The control diet followed the same procedure for BacD but it has only MRS broth in place of bacterial isolates, resulting in the antibiotics-contained diet. Sample sizes (n) and *P* values are in each figure.

food for 2–3 days to remove any resident commensal bacteria in their gut. These flies were then given fresh BacD every 2–4 days to measure their lifespan (Fig 1C). First, we found that the conditioned diet with *A. persici* Ai or *Gluconobacter sp.* Gdi significantly shortened the female lifespan of an outbred strain $w^{Dah}$, while that with *L. plantarum* Lsi or *Leuconostoc* Leui did not have such a drastic effect (Fig 1D). This was also the case for male flies and the other wild-type strain Canton-S (Figs 1E, S1B, and S1C), indicating a robust phenotype.

To test whether an *A. persici* Ai-conditioned diet promotes ageing, we measured the age-related pathological phenotypes. Accumulating evidence has demonstrated that intestinal stem cells (ISCs) are hyperproliferative in the aged gut, which reflects dysregulation of intestinal homeostasis [10,24]. The increased number of phosphorylated histone H3 (PH3)-positive cells in the aged gut suggested that *A. persici* Ai induced ISC hyperproliferation (Fig 1F). In contrast, the *L. plantarum* Lsi-conditioned diet had a milder effect (Fig 1F). We also found that ISC proliferation was promoted by *A. persici* Ai, not only during ageing but also in the young gut in response to bleomycin, a well-known inducer of DNA damage (S1D and S1E Fig). These data suggested that ISC activity is upregulated by *A. persici* Ai, which may lead to premature ageing of the gut. Increased ISC proliferation and a shortened lifespan by *Acetobacter spp.* were also observed in the previous literature [18,22], which further confirmed that BacD could recapitulate at least some of the phenotypes seen in response to live bacteria under a normal laboratory environment.

The negative geotaxis assay is widely used to quantify the healthspan of flies [25]. We used male flies to exclude the effect of eggs inside the female body, which affects their climbing speed. As we expected, chronic feeding of the *A. persici* Ai-conditioned diet but not the *L. plantarum* Lsi-conditioned diet decreased the climbing ability of the aged (Day 35) flies (Fig 1G). This phenotype was not seen in the young (Day 6) flies (Fig 1G), implying that the bacteria-conditioned diet did not have an acute negative effect on the climbing ability. Together, these data suggested that bacterial products in *A. persici* Ai promote ageing and decrease the organismal healthspan in both males and females.

## *A. persici* Ai increased host resistance to paraquat and oral infection

Depletion of *Acetobacteraceae* from the gut microbiome increases the lifespan and starvation resistance but decreases paraquat resistance [26], suggesting that *Acetobacter* spp. also have a benefit for the host. Expectedly, the *A. persici* Ai-conditioned diet increased paraquat resistance in male and female flies (Figs 2A, 2B, S2A and S2B). It also had a negative effect on starvation stress in females (Fig 2C and 2D), but this effect was less apparent in males (S2C and S2D Fig).

We also tested whether BacD affects the host defence against infection with the pathogen *Pseudomonas entomophila*. Survival of both female and male flies against oral infection by *P. entomophila* was improved by *A. persici* Ai-conditioned diet (Figs 2E and S2E). Interestingly, *A. persici* Ai did not increase the survival against septic infection to the pathogen, at least in males (S2F Fig), which suggested that intestinal immunity is upregulated by the products of *A. persici* Ai. To distinguish whether increased survival against *P. entomophila* represents increased tolerance or resistance, we performed quantifications of CFUs and ISC proliferation following oral infection with *P. entomophila*. The data showed that there was no decrease in CFUs or increase in ISC proliferation in flies fed a diet conditioned with *A. persici* Ai (S2G and S2H Fig). Instead, we observed that these flies displayed a relatively mild level of ISC proliferation compared to other groups (S2H Fig), which might imply a milder epithelial damage. These results suggest that the increased survival against *P. entomophila* observed in flies with *A. persici* Ai-conditioned diet may be due to increased tolerance rather than resistance to pathogen infection.

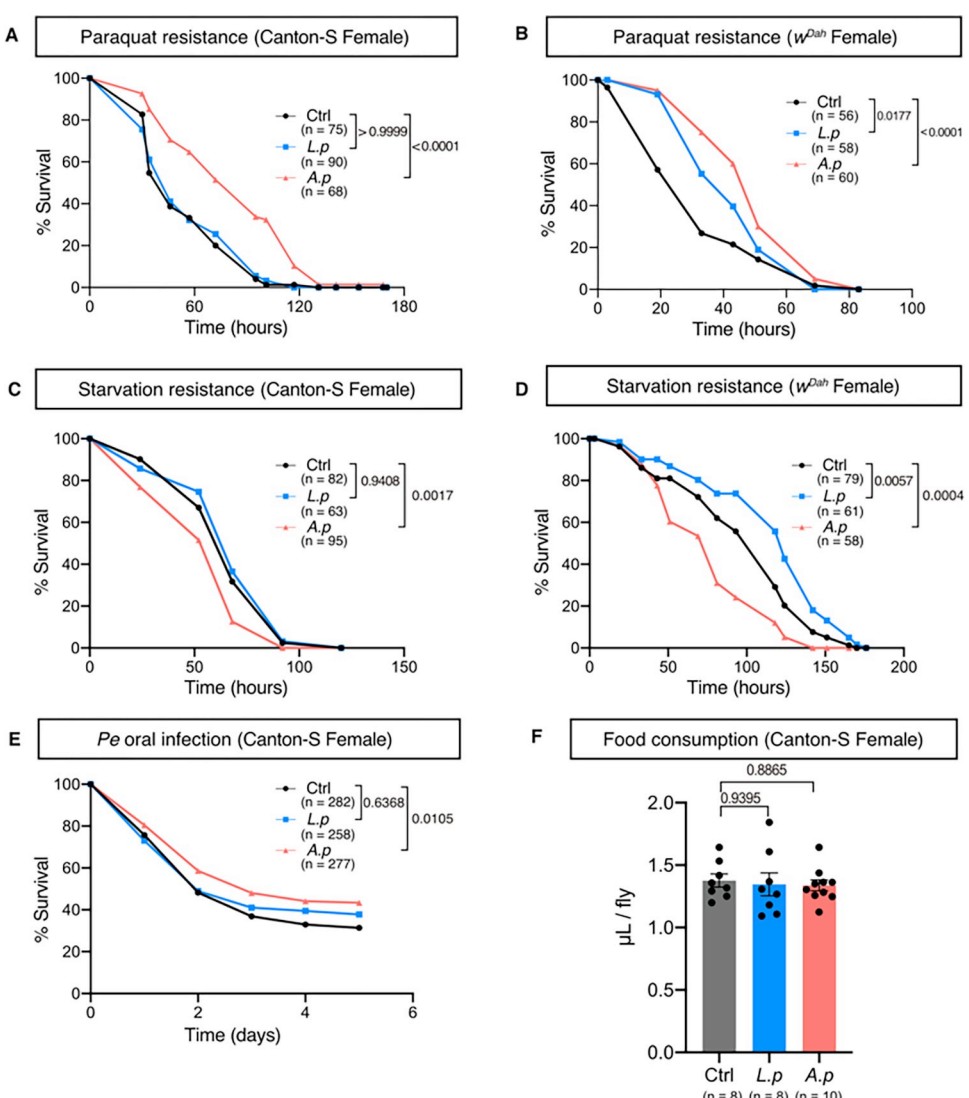

**Fig 2. *A. persici*-conditioned diet enhances resistance to paraquat and *Pe* oral infection but susceptibility to starvation.** (A)(B) Survival curve of female Canton-S (A) and $w^{Dah}$ (B) flies during 10 mM paraquat feeding after 5 days of a bacteria-conditioned diet (BacD). (C)(D) Survival curve of female Canton-S (C) and $w^{Dah}$ (D) flies during starvation stress after 5 days of BacD. (E) Survival curve of female Canton-S flies orally infected with *Pseudomonas entomophila (Pe)* after 5 days of BacD. A log-rank test was used to compare between control (Ctrl) and each BacD. (F) Food consumption of female flies by CAFE assay. Flies were treated with BacD for five days. Each point shows data of 7 or 8 flies. For the statistics, One-way ANOVA with Holm-Šídák's multiple comparison was used. The control diet followed the same procedure for BacD but it has only MRS broth in place of bacterial isolates, resulting in the antibiotics-contained diet. Sample sizes (n) and *P* values are in each figure.

## Bacteria-conditioned diet did not alter the feeding behaviour

Our data indicated that the feeding of BacD with *A. persici* Ai produced stronger phenotypes than that with L. plantarum Lsi. One hypothesis was that the flies consumed more *A. persici* Ai-conditioned diet because of the increased appetite, leading to a greater response. To evaluate this possibility, we conducted the capillary feeder assay (CAFE). This assay provides a measure of the flies' long-term feeding behaviour that quantifies the appetite of the flies. However, no differences were observed in feeding behaviour (Fig 2F). This finding suggests that BacD

does not influence the behaviour, and therefore, the differential effect of each BacD on the phenotype of the flies cannot be explained by the amount of food intake.

## *A. persici* Ai induced Imd-regulated antimicrobial peptides

To elucidate how *A. persici* Ai and *L. plantarum* Lsi distinctively impact the host, we investigated the transcriptomic response of the gut to each BacD. Midgut samples were collected after treating female $w^{Dah}$ flies with the conditioned diet for 24 hours (Fig 3A). As expected, we found that many genes and pathways upregulated by these BacDs were targets of the Imd pathway (S1 and S2 Tables). The Gene Ontology (GO) analysis of downregulated genes revealed that "Proteolysis" was significantly altered (FDR<0.01) in both *A. persici* Ai and *L. plantarum* Lsi (S2 Table), suggesting a potential role in the digestion of food. In contrast, the GO analysis of upregulated genes by *A. persici* Ai showed highly significant pathways related to carbohydrate metabolic processes such as glycolytic or pentose-phosphate pathway, as well as innate immune response or defense response. "Determination of adult lifespan" was found only in *A. persici* Ai (S2 Table). Interestingly, the GO analysis of upregulated genes in *L. plantarum* Lsi revealed that only two GO terms were below FDR<0.01 and these were related to amidase PGRPs (S2 Table). Our findings align with previous observations using live bacteria [18], as *A. persici* Ai was found to strongly induce antimicrobial peptides, while *L. plantarum* Lsi did so only mildly (Fig 3B) [18]. However, both *L. plantarum* Lsi and *A. persici* Ai sharply induced some of the Imd targets, such as *pirk* and amidase PGRPs (Fig 3B). These genes are known to negatively regulate the Imd pathway. We confirmed this result by quantitative RT–PCR using both Canton-S and $w^{Dah}$ female flies (Fig 3C and 3D). *PGRP-SC1a* was induced even more strongly in flies fed the *L. plantarum* Lsi-conditioned diet than in those fed the *A. persici* Ai-conditioned diet. To determine whether these genes were dependent on the Imd pathway, we utilized flies with mutations in *Dredd*, an essential component of the Imd pathway [26]. In the mutant $Dredd^{B118}$, the induction of *DptA* and *PGRP-SC1a* expression was completely abolished (Fig 3E). These data suggested that both *L. plantarum* Lsi and *A. persici* Ai have Imd activation capacities but distinct target gene spectra.

It has been shown that *L. plantarum* induced negative regulators of Imd, such as *PGRP-SCs*, in the copper cell region and posterior midgut but not the anterior midgut [27,28]. On the other hand, the induction of AMPs by infection was reported to be predominantly induced in the anterior midgut [27]. To investigate the region specificity of the Imd downstream gene expression pattern, we divided the gut into the anterior and posterior region and performed quantitative RT–PCR. For this experiment, the middle part of the gut containing the copper cell region was removed to avoid contamination of the other regions. We found that *DptA* induction by *A. persici* Ai was limited to the anterior region, while *PGRP-SC1a* expression induced by *L. plantarum* Lsi and *A. persici* Ai was limited to the posterior region (Fig 4A).

The homeobox gene *caudal* is known to be expressed in the posterior gut in larvae and it regulates immune-related genes [29]. We hypothesized that the region specificity of the expression of Imd target genes could be attributed to *caudal*, which is indeed expressed specifically in the posterior gut (Fig 4B). The gene expression of *caudal* was not affected by microbial association (Fig 4B). When we knocked down *caudal* using the gut genewitch driver $5966^{GS}$, we found that *DptA* induction was sharply increased, while *PGRP-SC1a* was suppressed in the gut (Fig 4C). Interestingly, *L. plantarum* Lsi was also capable of inducing *DptA* in the gut, although it did not reach statistical significance (Fig 4C). These data indicated that Caudal induced the *DptA-PGRP-SC1a* switch in the posterior gut. Together, our data imply that *A. persici* Ai can stimulate the Imd pathway in the whole midgut and induce *DptA* and *PGRP-SC1a*, while *L. plantarum* Lsi can do so only in the posterior midgut.

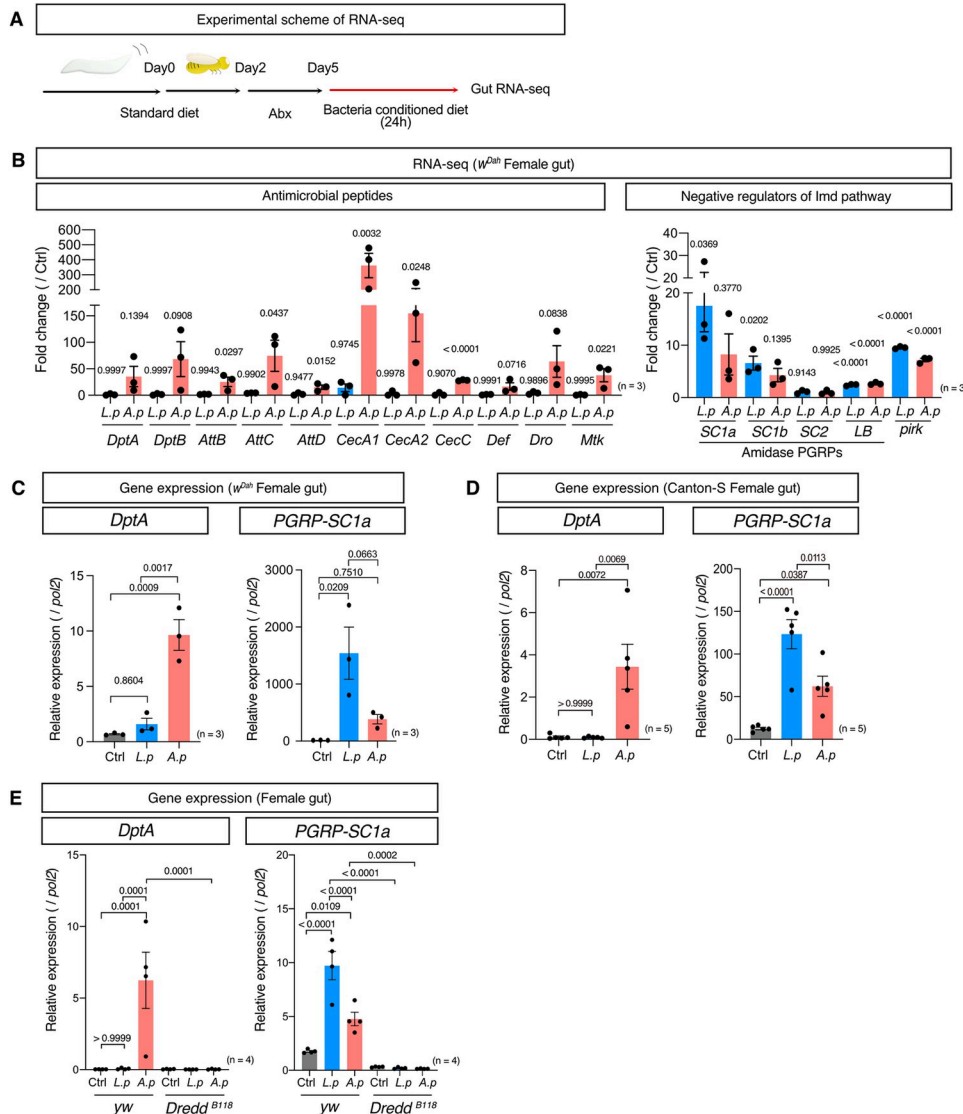

**Fig 3. Transcriptional regulation of distinct Imd target genes by bacterial species.** (A) The experimental overview of RNA-seq analysis of female $w^{Dah}$ gut after 24 hours of BacD. (B) RNA-seq read count data of antimicrobial peptide (AMP) genes and Imd negative regulator genes in $w^{Dah}$ female fly guts after 24 hours of BacD shown in fold change. For the statistics, ANOVA with Holm–Šídák's multiple comparison was used. (C)(D) Quantitative RT–PCR of *DptA* and *PGRP-SC1a* in $w^{Dah}$ (C) and Canton-S (D) female fly guts after 24 hours of BacD. For the statistics, ANOVA with Holm–Šídák's multiple comparison was used. (E) Quantitative RT–PCR of *DptA* and *PGRP-SC1a* in *yw* and *yw*-backcrossed $Dredd^{B118}$ female fly guts after 24 hours of BacD. For the statistics, ANOVA with Holm–Šídák's multiple comparison was used. The control diet followed the same procedure for BacD but it has only MRS broth in place of bacterial isolates, resulting in the antibiotics-contained diet. Sample sizes (n) and *P* values are in each figure.

## AMPs and amidase PGRPs were selectively induced via PGRP-LC and LE

The most interesting discovery was that only *A. persici* Ai induced AMP expression even though both *A. persici* Ai and *L. plantarum* Lsi have diaminopimelic (DAP)-type PGN. The Imd pathway is redundantly activated via two PGRPs, PGRP-LC and PGRP-LE. To investigate how these receptors contribute to the response to *L. plantarum* Lsi and *A. persici* Ai, we tested mutants of *Relish*, *PGRP-LC*, and *PGRP-LE* ($Rel^{E20}$, $PGRP-LC^{E12}$, $PGRP-LE^{112}$). As we

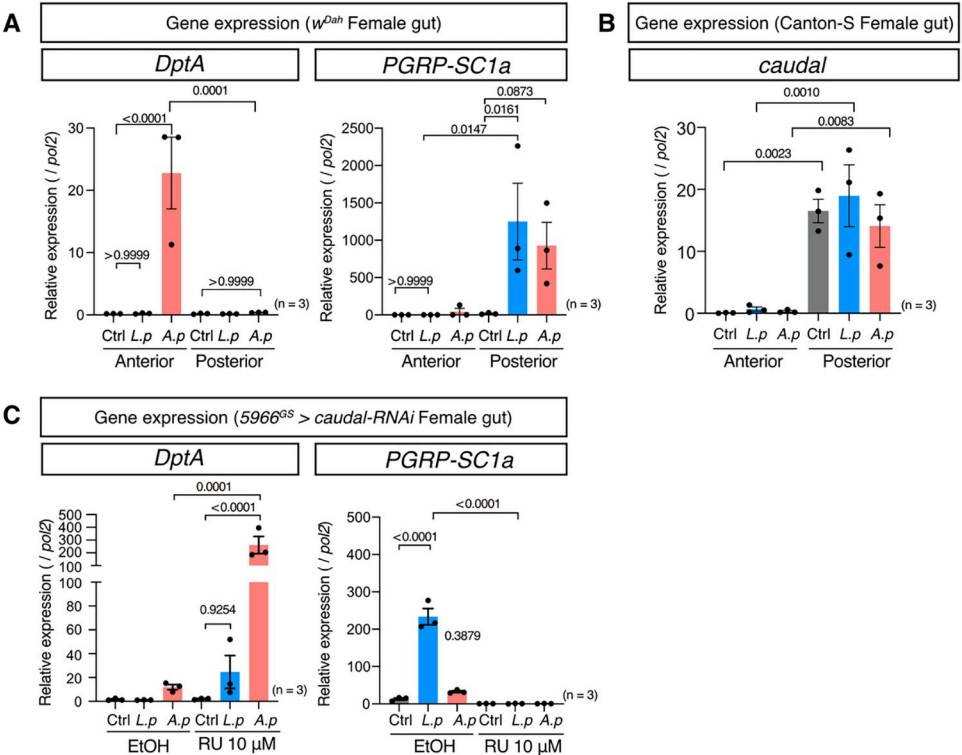

**Fig 4. Differential expression of Imd target genes in the anterior and posterior midgut.** (A) Quantitative RT–PCR of *DptA* and *PGRP-SC1a* (A) in female $w^{Dah}$ anterior and posterior gut after 24 hours of BacD. For the statistics, ANOVA with Holm-Šídák's multiple comparison was used. (B) Quantitative RT–PCR of *caudal* in female Canton-S anterior and posterior gut after 24 hours of BacD. For the statistics, ANOVA with Holm-Šídák's multiple comparison was used. (C) Quantitative RT–PCR of *DptA* and *PGRP-SC1a* in the gut of female $5966^{GS} > caudal$-RNAi flies after 24 hours of BacD. For the statistics, ANOVA with Holm-Šídák's multiple comparison was used. The control diet followed the same procedure for BacD but it has only MRS broth in place of bacterial isolates, resulted in the antibiotics-contained diet. Sample sizes (n) and P values are in each figure.

expected, both *DptA* and *PGRP-SC1a* were completely suppressed in $Rel^{E20}$ flies (Fig 5A and 5B). Intriguingly, *DptA* induction by *A. persici* Ai was completely suppressed in $PGRP-LC^{E12}$ flies. On the other hand, in $PGRP-LE^{112}$ flies, neither *L. plantarum* Lsi nor *A. persici* Ai induced *PGRP-SC1a* (Fig 5A and 5B), indicating that these two receptors regulate different target genes. *DptA* expression in *PGRP-LE* mutant flies was highly upregulated in the *A. persici* Ai-conditioned diet, which suggested that PGRP-LE-dependent induction of amidase PGRPs suppressed *DptA* induction. Interestingly, *PGRP-LE* was also expressed in the anterior gut (S3A Fig). Therefore, the differential expression of *DptA* and *PGRP-SC1a* in the anterior vs posterior gut is not due to the region-specific expression of PGRP-LC and LE. We also found that *DptA* was induced only by *A. persici* Ai in other tissues, such as the thorax and head, which was again suppressed in $PGRP-LC^{E12}$, but not $PGRP-LE^{112}$ (S3B and S3C Fig). Taken together, these data demonstrated that *A. persici* Ai systemically stimulates PGRP-LC, whereas *L. plantarum* Lsi only stimulates PGRP-LE in the posterior gut. Why *L. plantarum* Lsi does not activate the Imd pathway through PGRP-LE in the anterior gut is not clear; however, one can assume that the *L. plantarum* Lsi PGNs may be converted to "active" form (through modification, etc.) that allows it to be recognized by the receptor in the posterior region.

Since our BacDs includes the antibiotics cocktail, we checked whether side effect of antibiotics would interfere with the phenotypes caused by BacD. The differential induction of Imd

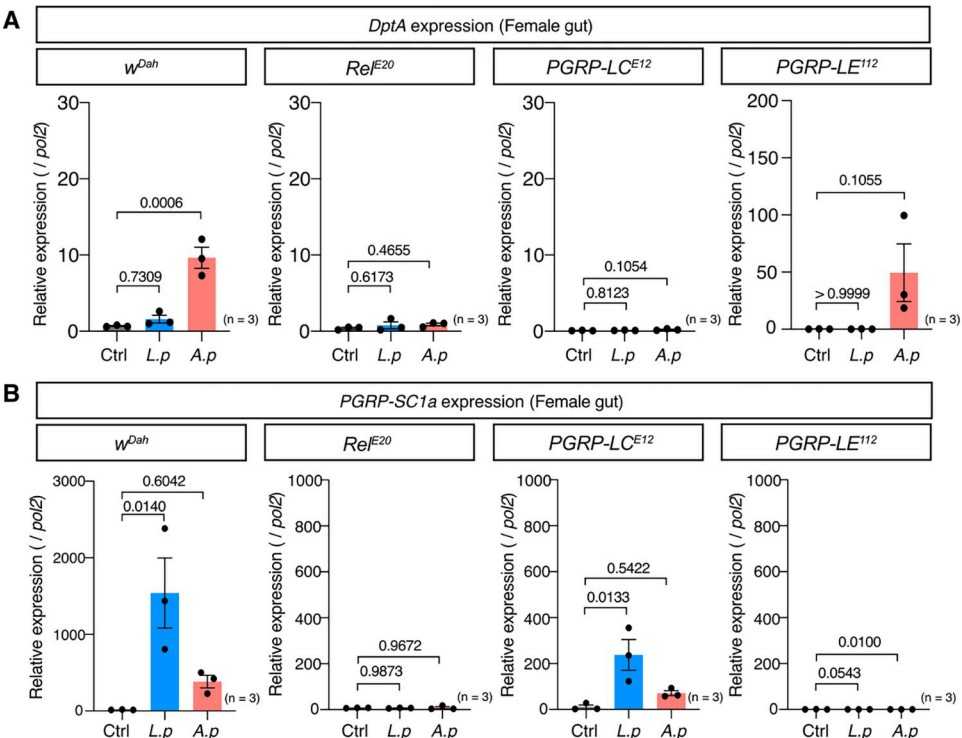

**Fig 5. Distinct receptor PGRPs regulate different Imd target genes in gut.** (A)(B) Quantitative RT–PCR of the genes *DptA* (A) and *PGRP-SC1a* (B) in female $w^{Dah}$, $Rel^{E20}$, $PGRP-LC^{E12}$, and $PGRP-LE^{112}$ fly guts after 24 hours of BacD. For the statistics, ANOVA with Holm-Šídák's multiple comparison was used. The control diet followed the same procedure for BacD but it has only MRS broth in place of bacterial isolates, resulted in the antibiotics-contained diet. Sample sizes (n) and *P* values are in each figure.

target genes as well as the enhanced paraquat resistance or the reduced starvation resistance with *A.persici* Ai-conditioned diet was observed even in the absence of the antibiotics cocktail (S4A, S4B and S4C Fig).

## Heat-killed *A. persici* Ai shortened lifespan

We hypothesized that the specific effects of *A. persici* Ai and *L. plantarum* Lsi on the host healthspan are due to their ability to stimulate distinct receptors for bacterial PGNs. If this is the case, changes in the level of metabolites in the conditioned diet (S1A Fig) are not important for determining lifespan, and possibly exposure to the bacterial cell wall is enough to induce the ageing phenotypes. Therefore, we simply fed the flies heat-killed bacteria (HK, Fig 6A).

We found that HK *A. persici* Ai, but not *L. plantarum* Lsi, at a dose of $OD_{600} = 40$ (this theoretically matches the quantity of BacD), shortened the lifespan in both sexes (Figs 6B, S5A and S5B). HK *A. persici* Ai also increased ISC proliferation in the aged gut (Figs 6C and S5C). As we expected, the induction pattern of Imd target genes by HK at a dose of $OD_{600} = 40$ in their guts resembled that caused by BacD (Fig 6D). HK *A. persici* Ai induced *DptA* even at the lowest dose ($OD_{600} = 0.4$) while *L. plantarum* Lsi could not do so even at the highest ($OD_{600} = 40$) (S5D and S5E Fig). In contrast, *L. plantarum* Lsi induced *PGRP-SC1a* expression significantly at the middle dose ($OD_{600} = 4$), while *A. persici* Ai boosted it only at the highest dose (S5D and S5E Fig). These data highlight the differential stimulation potential of PGRP receptors by each bacterial species.

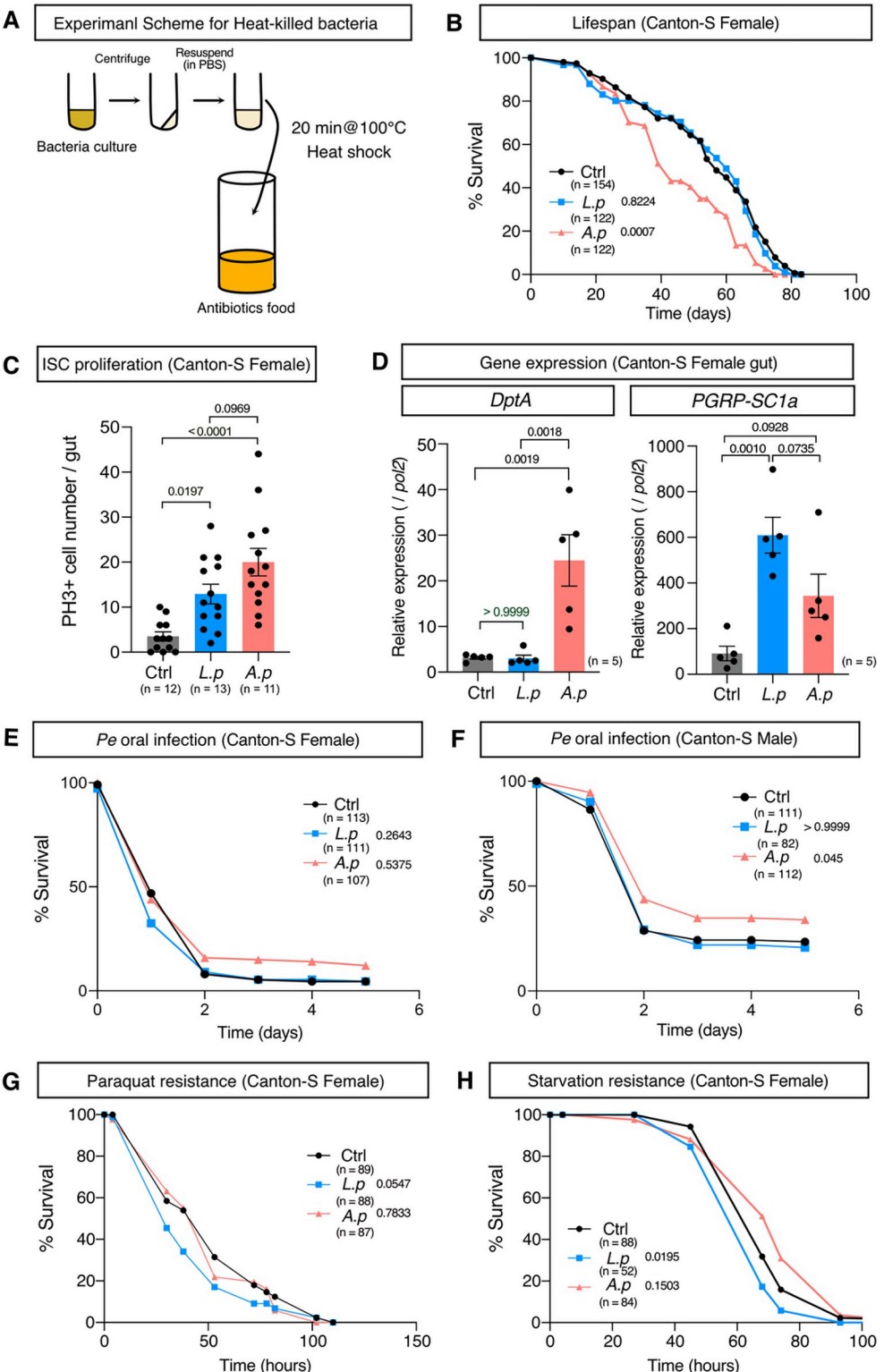

**Fig 6. Heat-killed *A. persici* shortens lifespan and increases the host defence but does not enhance paraquat resistance.** (A) An overview of the method to prepare the heat-killed bacteria (HK) diet. (B) Lifespan of female Canton-S flies fed the HK diet. A log-rank test was used to compare between control (Ctrl) and each HK diet. (C) Phospho-histone H3-positive cell numbers in the midgut of Canton-S female flies fed the HK diet for 40 days (Day45). For the statistics, ANOVA with Holm-Šídák's multiple comparison was used. (D) Quantitative RT–PCR of the genes

*DptA* and *PGRP-SC1a* in the female Canton-S fly gut after 24 hours of HK diet. For the statistics, ANOVA with Holm-Šídák's multiple comparison was used. (E)-(H) Resistance to oral infection with *Pseudomonas entomophila* (*Pe*) in female and male Canton-S flies fed the HK diet for five days (E, F), and resistance to 10 mM paraquat (G), or starvation (H) in female Canton-S flies with HK diet for five days in female Canton-S flies. A log-rank test was used to compare between control (Ctrl) and each HK diet. The control diet is the antibiotics-contained diet. Sample sizes (n) and *P* values are in each figure.

The host defence against oral infection with *P. entomophila* was mildly increased especially in males (Fig 6E and 6F). These data suggested that PGN of the *A. persici* Ai cell wall can recapitulate the effect of live bacteria or BacD. Interestingly, however, HK *A. persici* Ai did not increase the resistance to paraquat, nor decrease starvation resistance (Fig 6G and 6H). These data suggests that the stress resistance phenotypes require bacteria-derived metabolites or proteins. Therefore, the promotion of ageing and intestinal immunity is mechanistically independent of the altered stress resistance (S5F Fig).

## Purified PGNs induce distinct gene expression in the gut

To determine whether the differences in activity between heat-killed *A. persici* Ai and *L. plantarum* Lsi were solely attributed to the differences in PGNs rather than other components, we purified the PGNs from each bacterial species. As gram positive *L. plantarum* has wall teichoic acids (WTA) in the PGN layers, we also tested the effect of removing WTA on the phenotype. Strikingly, feeding with purified PGNs produced the same pattern of Imd target gene induction in the gut as BacD or HK (Fig 7A). Purified PGNs of *A.persici* Ai induced both *DptA* and *PGRP-SC1a* expression, while PGNs from *L. plantarum* Lsi induced only *PGRP-SC1a* expression in the gut. Removing WTA by HCl did not affect the gene expression pattern of *L. plantarum* Lsi at all. Furthermore, even a hundred-fold dilution of purified PGNs of *A.persici* Ai did not diminish the induction of *DptA* expression (Fig 7B). These results suggest that purified PGNs are sufficient for inducing the distinct gene expression patterns in the gut and the differences in PGN structure of two bacterial species are likely responsible for the differential stimulation of the receptors PGRP-LC/LE.

## *A. persici* Ai promotes ageing via intestinal PGRP-LC

Since *A. persici* Ai specifically induces PGRP-LC-dependent Imd activation and shortens lifespan, we asked whether host ageing was influenced by manipulating the Imd pathway. First, we used a progenitor-specific driver *esg^ts* to knock down *Rel*, *PGRP-LC*, and *PGRP-LE* to test whether age-related ISC proliferation is affected. We found that the *A. persici* Ai-conditioned diet failed to promote ISC proliferation when *Rel* or *PGRP-LC* was knocked down, but it did increase ISC proliferation when *PGRP-LE* was knocked down (Fig 8A). Therefore, stimulation of PGRP-LC by *A. persici* Ai autonomously increases ISC activity. Interestingly, PGRP-LC-knock down in enterocytes by *NP1^ts* also attenuated the ISC proliferation (S6A Fig), suggesting that enterocytes' PGRP-LC also contributed to promotion of age-related ISC proliferation by *A. persici* Ai.

Next, we investigated whether Imd activation in the gut was responsible for the lifespan reduction caused by *A. persici* Ai. Knockdown of *Rel* in enterocytes and enteroblasts using the drug-inducible *5966^GS* driver hampered the induction of Imd target genes in the gut by *A. persici* Ai-conditioned diet (Fig 8B). Under this condition, the lifespan was only mildly shortened, suggesting that intestinal Imd activation by *A. persici* Ai promotes ageing (Fig 8C). Knockdown of *PGRP-LC* using *5966^GS* mitigated the lifespan reduction by HK *A. persici* Ai from a 22% decrease in the median lifespan to a 17% decrease (Fig 8D). As the effect seemed milder than that of *Rel*-RNAi, we also used the conventional gut driver *NP1-Gal4*. In

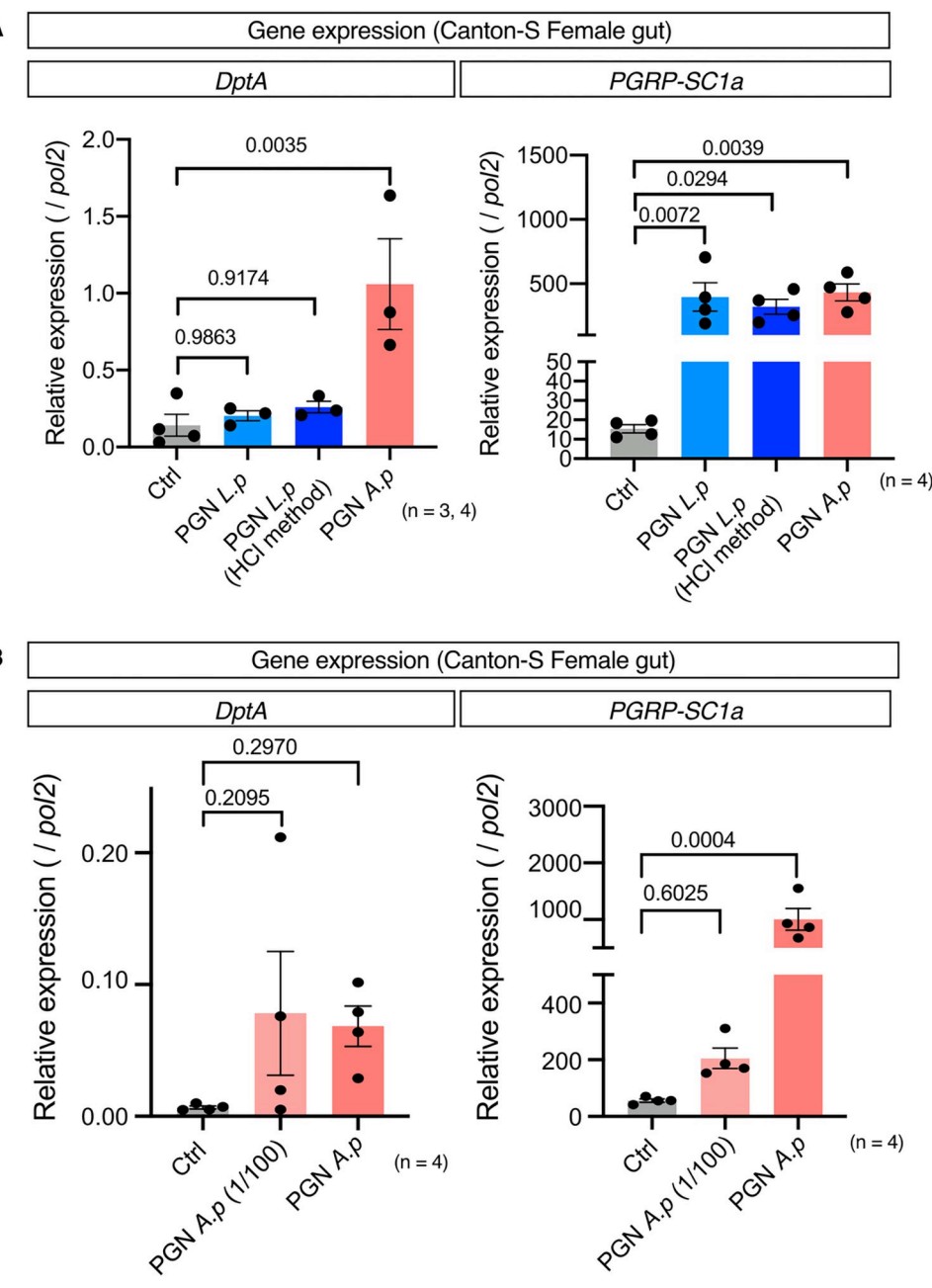

**Fig 7. Feeding *L. plantarum* Lsi and *A. persici* Ai purified peptidoglycans.** (A) Quantitative RT–PCR of the genes *DptA* and *PGRP-SC1a* in female fly Canton-S guts after 24 hours of purified peptidoglycan diet with the same concentration (measured and calculated by $OD_{254}$). For the statistics, ANOVA with Holm-Šídák's multiple comparison was used. (B) Quantitative RT–PCR of the genes *DptA* and *PGRP-SC1a* in female fly Canton-S guts after 24 hours of *A.persici* Ai purified peptidoglycan (1 or 100 times dilution) diet with (measured and calculated by $OD_{254}$). For the statistics, ANOVA with Holm-Šídák's multiple comparison was used. The control diet is the antibiotics-contained diet. Sample sizes (n) and *P* values are in each figure.

*NP1>PGRP-LC-RNAi*, HK *A. persici* Ai had a much smaller effect on lifespan than control *+>PGRP-LC-RNAi* flies (S6B Fig).

Since the increase of ISC proliferation caused by *A.persici* Ai was mitigated by knockdown of PGRP-LC in progenitor cells, we asked whether this also affected lifespan. Knockdown of

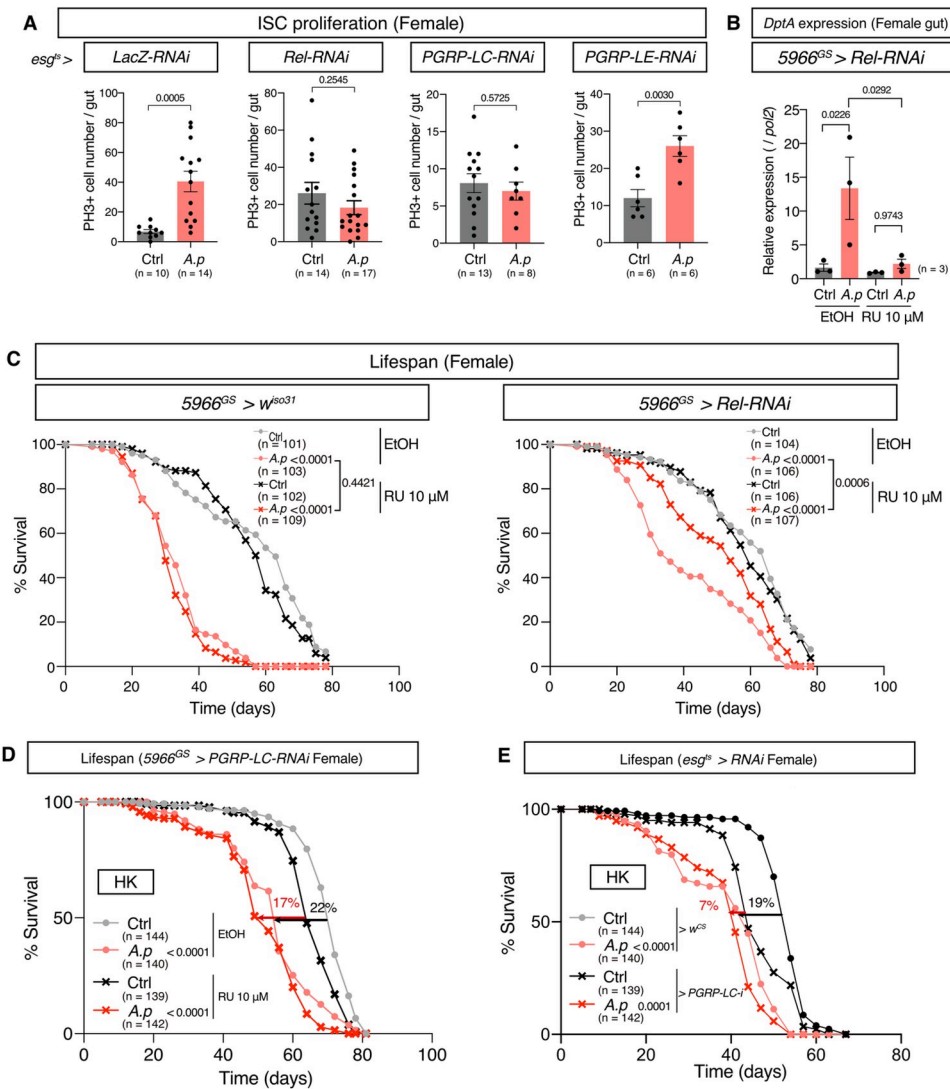

**Fig 8. PGRP-LC in the gut promotes ageing by *A. persici*.** (A) Phospho-histone H3-positive cell numbers in the midgut of 4-week-old female flies with BacD. The flies used were *Su(H)GBE-LacZ; esg-Gal4, tub-Gal80^ts^*, *UAS-GFP > LacZ-RNAi*, *Rel-RNAi*, *PGRP-LC-RNAi*, and *PGRP-LE-RNAi*, and the flies were reared at 30°C from their adult day2-4. Unpaired two-tailed Student's *t* test was used. (B) Quantitative RT–PCR of *DptA* in female *5966^GS^ > Rel-RNAi* fly gut after 24 hours of BacD with ethanol or 10 μM RU. For the statistics, ANOVA with Holm-Šídák's multiple comparison was used. (C) Lifespan of female *5966^GS^ > Rel-RNAi* flies treated with BacD and ethanol or 10 μM RU. A log-rank test was used to compare between control (Ctrl) and BacD with ethanol or 10μM RU. (D) Lifespan of female *5966^GS^ > PGRP-LC-RNAi* flies with heat-killed (HK) diet and ethanol or 10 μM RU. A log-rank test was used to compare between control (Ctrl) and HK diet with ethanol or 10μM RU. (E) Lifespan of female *esg^ts^ > w^CS^* and *esg^ts^ > PGRP-LC-RNAi* flies with HK diet. Flies were reared at 30°C during their adult period. A log-rank test was used to compare between control (Ctrl) and HK diet in each genotype. The 50% survival rate is compared in (D) and (E). The control diet in (A-C) followed the same procedure for BacD but it has only MRS broth in place of bacterial isolates, resulting in the antibiotics-contained diet. The control diet in (D) and (E) is the antibiotics-contained diet. Sample sizes (n) and *P* values are in each figure.

PGRP-LC using *esg^ts^* mitigated lifespan reduction by HK *A. persici* Ai from a 19% decrease in the median lifespan to a 7% decrease (Fig 8E). In contrast, *PGRP-LC*-RNAi using *Uro^GS^* (a Malpighian tubule driver), *elav^GS^* (a neuronal driver), and *WBFB^GS^* (a fat body driver) did not influence the lifespan shortening at all (S6C–S6E Fig). These data suggested that *A. persici* Ai

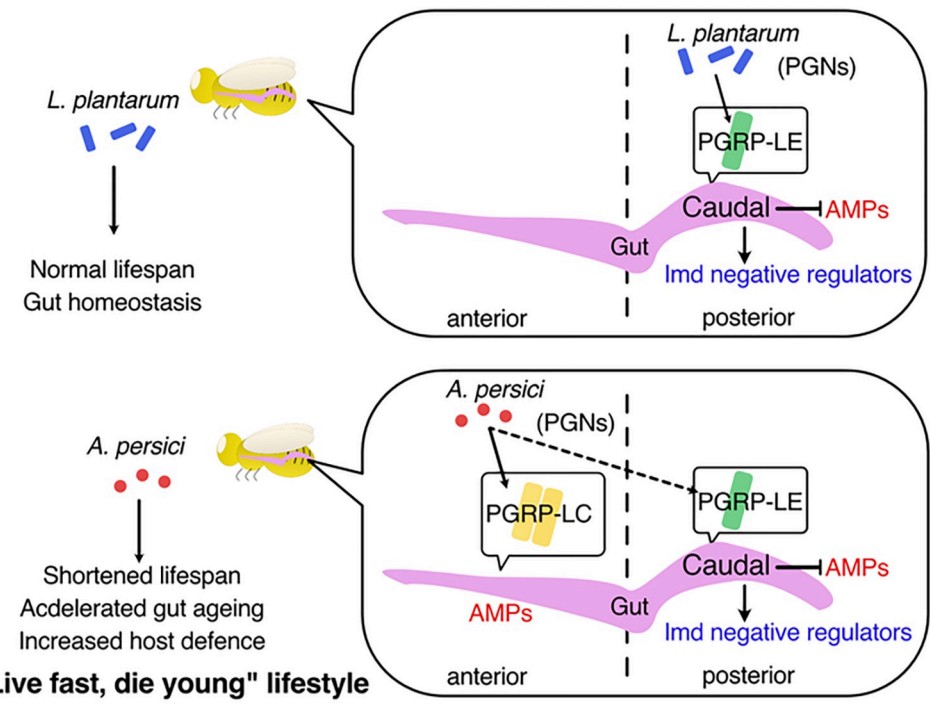

**Fig 9. Graphical summary of this study.**

promotes ageing, at least in part, through direct stimulation of the PGRP-LC receptor in both enterocytes and progenitors in the gut.

On the other hand, knockdown of *PGRP-LC* by *5966^GS* suppressed the enhanced resistance to paraquat by *A. persici* Ai-conditioned diet, suggesting that PGRP-LC-dependent Imd activation in gut enhances paraquat resistance (S6F Fig). Taken together, we concluded that sensing of the PGNs through the specific receptor mediates the shifts in the tradeoffs between the host defence capacity and lifespan (Fig 9).

## Discussion

In this study, we found that BacD with *A. persici* Ai, but not *L. plantarum* Lsi, upregulated AMP expression, promoted ISC proliferation, and shortened fly lifespan. This is consistent with our previous findings using live bacteria [18,22], suggesting that this type of conditioned diet can recapitulate gnotobiotic flies. The preparation of BacD is technically feasible under standard laboratory conditions and is potentially highly reproducible. Inactivation of the conditioning by antibiotics after 24 hours of bacterial inoculation prevents further fermentation and contamination by other bacteria. Gnotobiotic analysis may sometimes produce an artefact, as we cannot control how much of each bacterial species grows in the gut, which is affected by the host condition and genetic background. Especially when comparing two or more bacterial species, the quantity of the gut bacteria should be carefully considered. The mechanism by which the gut microbiota shortens the *Drosophila* lifespan is controversial and likely depends on the context, such as the microbial composition (both in quantity and quality), the dietary composition, and the host genotype [30,31]. Therefore, our BacD might be an alternative, accessible tool to quantitatively test how (an equal amount of) each bacterial species or strain impacts the host healthspan.

It is generally believed that bacteria need to be alive to be effective. This is why probiotics and prebiotics have flourished. However, recently, the significant impact of dead bacteria represented by those in pasteurized fermented foods has been frequently discussed, raising the possibility of "postbiotics" [32]. Postbiotics contain inactivated microbial cells or cell components that can benefit host health. Our BacD is useful for studying the mechanistic basis of postbiotics using isolated bacterial species.

Both PGRP-LC and PGRP-LE can recognize diaminopimelic (DAP)-type peptidoglycans from Gram-negative bacteria and some *bacilli* [33]. It was totally unexpected for us to find that PGNs from *A. persici* Ai stimulates PGRP-LC to induce AMPs in the anterior gut and to activate PGRP-LE for the Imd negative regulators in the posterior gut. During oral infection with the frequently used Gram-negative pathogen Ecc15, both PGRP-LC and LE contribute to inducing AMPs in the gut [16,27]. *Dpt* reporters are induced in several regions, such as the proventriculus, copper cells, and posterior midgut, upon Ecc15 ingestion [27,34]. *Dpt* upregulation in the proventriculus, the most anterior part of the gut, was found to be PGRP-LC dependent, while that in the posterior part was largely PGRP-LE-dependent in both the larval and adult gut [27]. PGRP-LC-dependent *Dpt-lacZ* in the anterior gut has also been reported during ingestion of Gram-negative, but not Gram-positive, PGN [35]. In our study, the *L. plantarum* Lsi-conditioned diet has very little effect on AMP induction but strongly induces negative regulators from the posterior gut via PGRP-LE, which is consistent with previous reports using monoassociation [12,27]. This mechanism is thought to involve immune tolerance to beneficial microbes in the gut. Therefore, we speculate that *A. persici* Ai possesses mixed characteristics of pathogenic and beneficial bacteria, as it can stimulate both LC-dependent AMPs and LE-dependent negative regulators at the same time. In this context, it is intriguing that *Acetobacter* spp., but not *L. plantarum* or *L. brevis*, has been shown to contribute to a protective priming mechanism for antiviral immunity in the adult gut [36]. This bacteria-specific effect is also due to the stimulation of PGRP-LC, which leads to the induction of the antiviral immune effector pvf2.

Our data suggested that two PGRP receptors distinguish the differences in the peptidoglycan structure between the two bacterial species. The sugar chain of these peptidoglycans consists of N-acetyl-D-glucosamine (GlcNAc) and N-acetylmuramic acid (MurNAc) repeats. The terminus of these sugar chains has a unique internal 1,6-anhydrobind in Gram-negative bacteria but not in *Bacilli* [37]. From their genome sequences, we found the genes responsible for anhydro-MurNAc formation in *A. persici* Ai and those responsible for *O*-acetylation of MurNAc and GlcNAc in *L. plantarum* Lsi (Table 1), suggesting that these two strains produce PGN with different modifications, as observed in related *Acetobacter* spp. and *L. plantarum*

**Table 1. Genes related to peptidoglycan modifications in the bacterial genome.**

| Modification | Gene name | KEGG Orthology | *A. persici* Ai | *L. plantarum* Lsi |
|---|---|---|---|---|
| GlcNAc N-deacetylation | *pgdA* | K22278 | -[*1] | -[*1] |
| MurNAc N-deacetylation | *pdaA* | K01567 | - | - |
| MurNAc N-glycolylation | *namH* | K14952 | - | - |
| MurNAc O-acetylation | *oatA/pat/pac* | NA | - | + |
| GlcNAc O-acetylation | *oatB* | NA | - | + |
| O-deacetylation | *ape* | NA | - | - |
| Anhydro-MurNAc | lytic transglycosylase | Several | + | -[*2] |

[*1], Carbohydrate esterase family 4 (CE4) exists in the genome.

[*2], mltG-related gene exists in the genome.

[38,39]. Due to the difficulty in obtaining a large amount of purified PGN, we measured only expression levels of the Imd target genes in the gut as a primary response of flies. However, given that ageing phenotypes are mediated by a "peptidoglycan"-recognition protein PGRP-LC, and caused by heat-killed *A. persici* Ai, we can logically attribute these phenotypes to the PGNs. Further biochemical and genetic analyses are needed to understand how specific PGN structure is recognized by the two PGRP receptors.

The stimulation of PGRP-LC by *A. persici* Ai leads to increased ISC proliferation. This result is consistent with a study showing that age-dependent ISC overproliferation was rescued by inactivating the Imd pathway [40]. Single-cell RNA-seq in the gut clarified that *PGRP-LC* was strongly expressed in progenitor cells, while *PGRP-LE* expression was higher in enterocytes [41]. This might be why ISC overproliferation was dependent on PGRP-LC, which was stimulated by *A. persici* Ai PGN. However, since knockdown of PGRP-LC in enterocytes also rescued the ISC hyperproliferation, there might be an additional mechanism affecting ISC proliferation in response to gut damage [42].

The cause of the shortened lifespan induced by heat-killed *A. persici* Ai is not completely understood. Our data suggest that suppressing immune activation by PGRP-LC only in enterocytes is sufficient to mitigate lifespan reduction. Thus, it is possible that PGRP-LC-dependent AMP production in the anterior gut leads to a shortened lifespan by disrupting gut homeostasis. We cannot deny the possibility that decreasing PGRP-LC in the gut can also decrease systemic Imd activation, as there might be a relay mechanism [43]. AMPs may directly cause cellular damage in aged animals. Excessive AMP expression would also increase transcriptional or translational stress (e.g., endoplasmic reticulum stress). Other Imd target genes can also promote ageing, such as GATAe, which regulates enterocyte shedding [34]. The exact mechanism of how chronic intestinal immune activation by *A. persici* Ai PGN leads to ageing remains to be clarified.

The gut microbiota can provide nutrients and stimulate nutrient-sensing mechanisms, which manifest when dietary conditions are suboptimal. In this study, we used a nutrient-rich standard fly diet; therefore, the benefit from the nutrient supply was minimal. However, the *A. persici* Ai-conditioned diet increased survival against paraquat or *P. entomophila* oral infection at the cost of lifespan. Interestingly, the shortened lifespan and increased paraquat resistance were mechanistically uncoupled, as feeding heat-killed *A. persici* Ai was sufficient for the former phenotype but not the latter. This fact suggests that the increased resistance requires *A. persici* Ai-produced metabolites, which have yet to be identified. *Drosophila* would take advantage of this beneficial metabolite from *A. persici* Ai and make a trade-off decision to accept the risk of accelerated ageing. Thus, *A. persici* Ai can render the flies "Futoku-Mijikaku (Thick and Short)" or the so-called *live fast*, *die young* lifestyle (Fig 9).

## Materials and methods

### Fly stocks, husbandry, and lifespan

*D. melanogaster* stocks were raised on a standard yeast-cornmeal diet containing 8 g agar, 60 g glucose, 40–45 g corn flour, 40–60 g yeast, 4 mL propionic acid, and 6–15 mL 10% butyl p-hydroxybenzoate per litre. Adult flies were maintained for two days after eclosion for maturation and mating on the standard diet. The flies were then sorted by sex and maintained at a fixed density of 30 flies/vial under conditions of 25°C and 60–65% humidity with 12 h/12 h light/dark cycles. Development density was controlled by adding a fixed volume of embryos to the standard diet to avoid overcrowding.

The flies were fed antibiotic diet for two to three days to deplete resident bacteria. The antibiotic diet was prepared by mixing rifampicin 50–200 mg/L (Wako 185–01003), tetracycline

12.5–50 mg/L (Wako 203–08592), and ampicillin 125–500 mg/L (Wako 203–08592) into the conventional diet. To activate the gene-switch system, 10 μM RU486 (Tokyo-Kasei, M1732) was added to the standard fly diet before bacterial conditioning. To measure their lifespan, the number of dead flies was counted every two to four days when the flies were transferred to fresh vials. Canton-S, wDah, wCS (Canton-S backcrossed to wiso31 eight times), wiso31, and yw flies were used as control strains. Other fly lines used in this study were DreddB118 [26], 5966GS, UAS-GFP [44], WBFBGS(S106+S32)[45], UroGS [46], Su(H)GBE-LacZ, esgts, NP1-Gal4 [47], UAS-caudal-RNAi (VDRC, v3361), UAS-Rel-RNAi (BDSC, 33661), PGRP-LC-RNAi (BDSC, 33383), and UAS-PGRP-LE-RNAi (BDSC, 60038). The RelE20, PGRP-LCE12, and PGRP-LE112 lines have been described previously [17,48]. NP1-Gal4 was backcrossed to wiso31 eight times.

## Bacterial stocks and isolation

*A. persici* Ai, *L. plantarum* Lsi, and *Gluconobacter* Gdi were isolated previously [23] *Leuconostoc* sp. Leui was isolated from the gut of $da^{GS}$ flies maintained in the laboratory. The gut homogenate was smeared on an MRS agar (Kanto Chemical, C711361-5) plate and incubated for two days at 30˚C. A single colony was picked, and the bacterial genus was identified from the 16S rRNA sequence. Blast analysis showed that the 16S rRNA sequence of *Leuconostoc* sp. Leui has 99% identity with the *Leuconostoc mesenteroides* or *L. pseudomesenteroides* strains.

## Bacteria-conditioned diet

The bacteria-conditioned diet (BacD) was prepared using isolated bacterial strains. Freeze stock of the bacterial isolates was precultured in 3–5 mL MRS broth (Kanto Chemical, 711359–5) overnight. This preculture was then transferred to 20–40 mL of MRS broth to culture the strain for 12–16 hours. The absorbance ($OD_{600}$) of the culture medium was measured by a DEN-600 Photometer (Funakoshi, BS-050109-AAK). Subsequently, a standard fly diet was inoculated with 100 μL of bacterial culture (or just MRS broth for the negative control) at $OD_{600}$ = 0.2 or 0.4 and incubated at 25˚C for 24 hours. We chose $OD_{600}$ = 0.2 or 0.4 as it produces comparable level of CFU in the standard diet after keeping wild type flies for three days. For the CFU count of the diet, 50-200mg of the diet surface was collected by inserting a straw and transferring the samples into 100μL of PBS. The samples were homogenized and then plated on MRS-agar plates using EddyJet2. Finally, 100–120 μL of antibiotic cocktail (rifampicin 10 g/L (Wako 185–01003), tetracycline 2.5 g/L (Wako 203–08592), and ampicillin 12.5 g/L (Wako 203–08592) in 100mM HCl) was added to each diet (including the negative control) to prevent further conditioning. The dosage of antibiotics was optimised based on the viability of bacteria in BacD. The absence of live bacteria in BacD was periodically checked by plating a piece of BacD to MRS agar. The conditioned diet was dried at room temperature, stored at 4˚C, and used within two weeks of preparation.

## Metabolite analysis of the Bacteria-conditioned diet

Metabolite are measured by LC-MS/MS as reported previously [18,49]. Briefly, approximately 100 mg of fly diet at/near the surface was picked up by a plastic straw and homogenized in 150 μL of 80% methanol containing 10 μM of internal standards (methionine sulfone and 2-morpholinoethanesulfonic acid), and deproteinized by mixing with 75 μL of acetonitrile. The supernatant was applied into a 10-kDa centrifugal device (Pall, OD010C35) and the flow-through was evaporated completely using a centrifugal concentrator (TOMY, CC-105). The samples were resolubilised in Ultrapure water and injected to the LC-MS/MS with a PFPP column (Discovery HS F5 (2.1 mm × 150 mm, 3 μm)). The MRM parameters for ~150

metabolites were optimized by the injection of the standard solution, through peak integration and parameter optimization with the software (Labsolutions, Shimadzu). The analysis of the fly diet quantified 60 metabolites. These data were visualised by MetaboAnalyst 5.0.

## Intestinal stem cell proliferation

Guts from female flies were dissected in PBS. The tissues were fixed with 4% paraformaldehyde for 30 to 60 min and washed with PBS-T (0.1% Triton X-100) three times (10 min each time), followed by blocking with 4% normal donkey serum in PBS-T (NDS-T) for 30 to 60 min. The tissues were stained overnight with anti phospho-histone H3 (Abcam, ab10543, 1:2000 dilution) antibody. Samples were washed with PBS-T three times (flush×1 and 10 min×2) and stained with Donkey anti-rat IgG Alexa Fluor 488 (Thermo Fisher Scientific, A21208, 1:1000 dilution) and Hoechst 33342 (Thermo Fisher Scientific, 1:1000 dilution) for two hours at room temperature. The number of phospho-histone H3-positive cells in the whole gut was counted manually under a fluorescence microscope.

For quantification of ISC proliferation upon *P. entomophila*, female flies were collected 16 hours after beginning the oral infection. These flies were dissected into PBS and the guts were immediately fixed with 4% paraformaldehyde in PBS for 30 min at room temperature. For the guts, the samples were rinsed three times in 0.2% Triton X-100 in PBS (0.2% PBT), rinse sequentially with 30%, 50%, 70%, 90% EtOH in pure water and incubated for 10 min with 90% EtOH. Then the guts were rinsed two times in 0.2% PBT and blocked with 2% BSA in 0.2% PBT (blocking solution) for 1 h. The samples were incubated with Phospho-Histone H3 (Ser10) Antibody (Cell Signaling Technology, 9701S) in blocking solution (1:500) overnight at 4°C. The guts were rinsed three times with 0.2% PBT for 15 min, incubated with Goat anti-Rabbit IgG (H+L) Highly Cross-Adsorbed Secondary Antibody, Alexa Fluor Plus 555 (Invitrogen, A32732) in blocking solution (1:500) for two hours, washed with 0.2% PBT and nuclei were stained with 4', 6-diamidino-2-phenylindole (DAPI; 3 μg ml−1, Dojindo, Japan). The samples were then rinsed two times with 0.2% PBT and mounted in Fluorsave (FluorSave Reagent 345789, Merck). The samples were visualized with a conventional fluorescent microscope (BX53, OLYMPUS)

## Climbing assay

A negative geotaxis assay was used to quantify climbing ability. For each condition, 10–30 flies were transferred to each cuboid vial (w: 2.6 cm, d: 1 cm, h: 16.4 cm) using a funnel. These vials were placed in front of a light box (HOZAN, 8015–011902) inside a dark tent to shed light from behind. Flies inside were dropped to the bottom of the vial by three consecutive taps. A minute after this practice tap, the actual assay was performed and recorded using a web camera (Logicool, 860–000336). The height of the flies at 10 sec after the tap was measured manually.

## Starvation/paraquat resistance

Adult flies were transferred to 1% agar (Wako, 010–15815) to detect starvation resistance or to a 1% agar (Wako, 010–15815), 5% sucrose (Wako, 196–00015), 10 mM paraquat (1,1'-dimethyl-4,4'-bipyridine-1,1'-diium dichloride) (TCI, D3685) diet to detect paraquat resistance. The number of dead flies was counted 1–3 times per day until all flies were dead.

## Septic and oral infection of *Pseudomonas entomophila*

Flies were infected with *P. entomophila* wild-type strain L48 (provided by Dr. B. Lemaitre) by pricking (septic infection) or by feeding (oral infection). Oral infection was performed as

previously demonstrated [50]. *P. entomophila* was incubated in LB medium at 29˚C overnight, and the bacterial pellet was collected by centrifugation. The microbe solution was obtained by mixing a pellet of *P. entomophila* with a culture supernatant (1:1), resulting in the concentration of the bacteria was approximately $OD_{600} = 200$. Adult flies were kept at 29˚C for two hours in an empty vial for starvation and then transferred to a vial containing the *P. entomophila* culture. The *P. entomophila* pellet was mixed with an equal amount of the culture supernatant and soaked into a filter paper disc that completely covered the surface of the standard fly diet. Flies were held in this diet at 29˚C, and mortality was monitored for five days.

## CFU count for *P. entomophila* infection

Flies were collected 6 hours after post oral infection with *P. entomophila*. Five flies were washed with 70% ethanol and then with PBS, and transferred to 2-mL screw tubes (SARSTEDT, 72.693). The flies were homogenised with Precellys 24 (Bertin Technologies) in 500 μL of LB medium and 100 μL of glass beads at 6,000 rpm for 30 s. The homogenates were plated onto LB-agar plates and cultured at 29˚C for 2 days. The microbe counting (CFU assays) were performed as previously described [51].

## Feeding behaviour

For CAFE assay, capillaries (9600105) were filled with a solution containing 2mg/mL Acid red 52 (Wako, 018–10012) and 5% sucrose (Wako, 196–00015) by simply dipping it into the solution. A customized cap with 4 wholes slightly larger than the bottom of the capillary was prepared. Capillaries filled with the solution were poked in the wholes until it slightly stuck out from the back side (2-3mm, make sure flies can suck the solution), and vials containing 6–8 flies were capped with this. Vials were not erected until the assay began. Right after the assay began, the liquid surface of the capillaries was recorded by marking them with a pen. Vials were collected 24 hours later, and the liquid surface was marked with another pen with a different color. Finally, the distance between the initial mark and the second mark was measured with a ruler. As a control, a vial without flies were also prepared, which represented the evaporation of the solution. Finally, after evaporation was considered, the net length was converted to the amount of food consumption.

## Transcriptomic analysis and quantitative RT–PCR

For transcriptomic analysis, 3' RNA-seq analysis was performed as previously described [49]. Briefly, the guts of $w^{Dah}$ flies were dissected. Crop and Malpighian tubules were carefully removed. Triplicate samples were prepared for each experimental group. Total RNA was purified using a ReliaPrep RNA Tissue Miniprep kit (Promega, z6112). The RNA was sent to Kazusa Genome Technologies to perform the library preparation and sequencing. Raw reads were analysed by the BlueBee Platform (LEXOGEN), which performs trimming, alignment to the *Drosophila* genome, and counting of the reads. The count data were statistically analysed by the Wald test using DESeq2. The result has been deposited in DDBJ under the accession number DRA015054.

For quantitative RT–PCR analysis, total RNA was purified from 3–4 tissues of female flies using a ReliaPrep RNA Tissue Miniprep kit (Promega, z6112). Quantitative RT-PCR was performed using a OneTaq RT–PCR kit (Promega, M0482S) with qTOWER3 (Analytik Jena), or PrimeScript RT reagent Kit (Takara, RR037A) and TB Green Premix Ex Taq (Tli RNaseH Plus) (Takara, RR820W) with Quantstudio6 Flex Real Time PCR system (ThermoFisher). The primer sequences are listed in Table 2.

**Table 2. Sequences of primers.**

| Forward | Sequence | Reverse | Sequence |
|---|---|---|---|
| RNA Pol2_F | CCTTCAGGAGTACGGCTATCATCT | RNA Pol2_R | CCAGGAAGACCTGAGCATTAATCT |
| DptA_F | CGTCGCCTTACTTTGCTGC | DptA_R | CCCTGAAGATTGAGTGGGTACTG |
| PGRP-SC1a_F | GCTCCGGCTACATCCTGTAC | PGRP-SC1a_R | TCGTTCCAGATGTGAGTGCC |
| PGRP-LC_F | AGGCCGTCACAGTTACAGTG | PGRP-LC_R | GTGGTGGCCAGTACGATACC |
| PGRP-LE_f | AGCACTATGACACTAGGCACT | PGRP-LE_R | GTCTGAATGCTGTTGATCGAGT |
| Caudal_F | CACAACCACAACCAGGCAAA | Caudal_R | AGGCGCTGGAAGTCGGTGTA |

## Heat-killed bacteria

The bacterial culture was centrifuged at 8000×g for 5 min to obtain a pellet. After measuring the $OD_{600}$, the pellet was resuspended in PBS to prepare a bacterial culture that theoretically matched an absorbance of $OD_{600} = 40$. After heat shock at 100°C for 15–20 min, 100 μL of the pellet sample was added to an antibiotic diet and dried at room temperature.

## Peptidoglycan purification

Peptidoglycan was purified and quantified as previously explained [52]. Briefly, microbes were cultured in a culture medium (MRS broth) for 24–48 hours until it reached $OD_{600} = 2.7$–$3.2$. The bacterial culture was then centrifuged at 5,000g for 10 minutes, and the pellet was collected and washed with PBS. The samples were suspended in 25 mM sodium phosphate (PB) and incubated in boiling SDS (8%), then centrifuged at 45,000g for 30 minutes. The pellet was washed with PB repeatedly 4–6 times until there was no more sign of SDS. For *L. plantarum* Lsi, the sample was then processed with 1N HCl for 4 hours at 37°C in a shaker. The samples were then incubated with 100 μg of α-amylase at 37°C for 1 hour, and then with 200 μg of pronase at 37°C for overnight. After boiling with SDS (4%) and washed several times with PB, the purified PGNs were finally suspended in 500μL of PB. The quantification of PGN solution were performed based on the absorbance at the wavelength $OD_{254}$ measured with the plate-reader (nivo). If we assume the efficiency of purification is 100%, the dose of purified PGN in the final solution theoretically matches that of approximately $OD_{600} = 2000$. A hundred-fold dilution of PGN of *A. persici* Ai leads to $OD_{600} = 20$.

## Statistics

Statistical analysis and graph drawing were performed using GraphPad Prism 7 or 8. For survival curves, OASIS 2 was used to perform the log-rank test to compare the curve of interest with the control [53]. The number of samples (n) for all experimental data indicate the biological replicates. Bar graphs are drawn as the mean ± s.e.m., with all the data points shown by dots. *P* values and sample numbers are indicated in each graph. All the data were reproduced at least twice.

## Supporting information

**S1 Fig. *A. persici*-conditioned diet alters metabolite composition, shortens lifespan in males, and increases ISC activity.** (A) The effect of bacterial conditioning on metabolite levels. (B)(C) Lifespan of male $w^{Dah}$ (B) and Canton-S (C) flies with BacD. A log-rank test was used to compare between control (Ctrl) and each BacD. (D) Experimental scheme of bleomycin treatment in the phospho-histone H3-positive cell number counting experiment. (E) Phospho-histone H3-positive cell numbers in the midgut of $w^{Dah}$ female flies after five days of

BacD and overnight treatment with 2.5 μg/mL bleomycin. The control diet followed the same procedure for BacD but it has only MRS broth in place of bacterial isolates, resulting in the antibiotics-contained diet. For the statistics, one-way ANOVA with Holm-Šídák's multiple comparison was used. Sample sizes (n) and *P* values are in each figure.
(TIFF)

**S2 Fig. *A. persici*-conditioned diet enhances resistance to paraquat and *Pseudomonas ento-mophila* (*Pe*) oral infection in male flies.** (A)(B) Survival curve of male Canton-S (A) and $w^{Dah}$ (B) flies during 10 mM paraquat feeding after 5 days of a bacteria-conditioned diet (BacD). A log-rank test was used to compare between control (Ctrl) and each BacD. (C)(D) Survival curve of male Canton-S (C) and $w^{Dah}$ (D) flies during starvation stress after 5 days of BacD. A log-rank test was used to compare between control (Ctrl) and each BacD. (E)(F) Survival curve of male Canton-S flies orally (E) or septically (F) infected with *Pseudomonas ento-mophila* (*Pe*) after 5 days of BacD. A log-rank test was used to compare between control (Ctrl) and each BacD. (G)(H) Colony forming units (CFUs) (G) and phospho-histone H3-positive cell numbers (H) in the midgut of female after 6 (G) or 16 (H) hours after oral infection with *P.entomophila*. Canton-S flies used were given BacD for 5 days before the assay. For the statistics, One-way ANOVA with Holm-Šídák's multiple comparison was used. The control diet followed the same procedure for BacD but it has only MRS broth in place of bacterial isolates, resulting in the antibiotics-contained diet. Sample sizes (n) and *P* values are in each figure.
(TIFF)

**S3 Fig. *A. persici*-conditioned diet induces *DptA* expression in the head and thorax via *PGRP-LC*.** (A) Quantitative RT–PCR of the genes *PGRP-LC* and *PGRP-LE* in female Canton-S anterior and posterior gut after 24 hours of BacD. One-way ANOVA with Holm-Šídák's multiple comparison was used. (B, C) Quantitative RT–PCR of *DptA* in female $w^{Dah}$, $Rel^{E20}$, $PGRP-LC^{E12}$, and $PGRP-LE^{112}$ fly heads (B) and thoraxes (C) after 24 hours of BacD. One-way ANOVA with Holm-Šídák's multiple comparison was used. The control diet followed the same procedure for BacD but it has only MRS broth in place of bacterial isolates, resulted in the antibiotics-contained diet. Sample sizes (n) and *P* values are in each figure.
(TIFF)

**S4 Fig. The effect of antibiotic cocktail on the phenotypes is minimal.** (A) Survival curve of female Canton-S flies during 10 mM paraquat feeding after 5 days of a bacteria-conditioned diet (BacD) without adding the antibiotics cocktail. A log-rank test was used to compare between control (Ctrl) and BacD without antibiotics. (B) Survival curve of female Canton-S flies during starvation stress after 5 days of BacD without adding the antibiotics cocktail. A log-rank test was used to compare between control (Ctrl) and BacD without antibiotics. (C) Quantitative RT–PCR of *DptA* in female Canton-S fly gut after 24 hours of BacD without adding the antibiotics cocktail. These experiments were conducted following the exact same scheme as BacD experiments, except that the BacD was prepared without the addition of the antibiotics cocktail. For the statistics, one-way ANOVA with Holm-Šídák's multiple comparison was used. The control diet has only MRS broth in place of bacterial isolates without the antibiotics cocktail. Sample sizes (n) and *P* values are in each figure.
(TIFF)

**S5 Fig. Heat-killed *A. persici* shortens lifespan in male flies, and the summary of pheno-types.** (A)(B) Lifespan of male Canton-S (A) and $w^{Dah}$ (B) flies with the heat-killed (HK) diet. A log-rank test was used to compare between control (Ctrl) and each HK diet. (C) Phospho-histone H3-positive cell numbers in the midgut of female $w^{Dah}$ flies fed the HK diet for 40 days. (D) (E) Quantitative RT–PCR of *DptA* and *PGRP-SC1a* after 24 hours of several dilutions

of HK diet with either *L.plantarum* (D) or *A.persici* (E). For the statistics, one-way ANOVA with Holm-Šídák's multiple comparison was used. (F) The summary of phenotypes in flies with BacD and HK diet. The control diet is the antibiotics-contained diet. Sample sizes (n) and *P* values are in each figure.
(TIFF)

**S6 Fig. *PGRP-LC* knockdown in gut mitigates the decrease in lifespan caused by heat-killed *A. persici*.** (A) Phospho-histone H3-positive cell numbers in the midgut of $NP1^{ts} >$ *PGRP-LC-RNAi* flies after 38days of heat-killed (HK) diet. Flies were reared at 30˚C during their adult period. For the statistics, one-way ANOVA with Holm-Šídák's multiple comparison was used. (B) Lifespan of female $w^{iso31} > PGRP\text{-}LC\text{-}RNAi$ and *NP1-Gal4 > PGRP-LC-RNAi* flies with HK diet. The 50% survival rate is compared. (C)-(E) Lifespan of female $Uro^{GS}$, *ElavGS*, and $WBFB^{GS} > PGRP\text{-}LC\text{-}RNAi$ flies with HK diet and ethanol or 10 μM RU. (F) Survival curve of female flies during 10 mM paraquat feeding after 5 days of BacD and ethanol or RU 10μM. A log-rank test was used to compare between control (Ctrl) and BacD with ethanol or 10μM RU. The control diet in (A)-(E) is the antibiotics-contained diet. The control diet in (F) followed the same procedure for BacD but it has only MRS broth in place of bacterial isolates, resulted in the antibiotics-contained diet. Sample sizes (n) and *P* values are in each figure.
(TIFF)

**S1 Table. Differentially expressed genes in the gut of flies with *A. persici* Ai- or *L. plantarum* Lsi-conditioned diet.**
(XLSX)

**S2 Table. Gene-ontology analysis of DEGs in the gut of flies with *A. persici* Ai- or *L. plantarum* Lsi-conditioned diet.**
(XLSX)

**S1 Data. All numerical data for the graphs and their statistics.**
(XLSX)

## Acknowledgments

We would like to acknowledge Miura's lab and Obata's lab members for the technical assistance and critical suggestions. We thank the National Institute of Genetics, Vienna Drosophila Resource Center, Bloomington Drosophila Stock Center, and Kyoto Drosophila Stock Center for fly stocks.

This manuscript was edited by one or more of the highly qualified native English speaking editors at AJE.

## Author Contributions

**Conceptualization:** Toshitaka Yamauchi, Masayuki Miura, Fumiaki Obata.

**Data curation:** Toshitaka Yamauchi, Hina Kosakamoto, Takumi Murakami, Hiroshi Mori.

**Formal analysis:** Taro Onuma, Toshitaka Yamauchi, Hina Kosakamoto, Hibiki Kadoguchi, Takayuki Kuraishi, Hiroshi Mori.

**Funding acquisition:** Takayuki Kuraishi, Masayuki Miura, Fumiaki Obata.

**Investigation:** Taro Onuma, Toshitaka Yamauchi, Hina Kosakamoto, Takumi Murakami, Hiroshi Mori, Fumiaki Obata.

**Methodology:** Taro Onuma, Toshitaka Yamauchi, Hina Kosakamoto, Hibiki Kadoguchi, Takayuki Kuraishi, Fumiaki Obata.

**Project administration:** Fumiaki Obata.

**Supervision:** Masayuki Miura, Fumiaki Obata.

**Visualization:** Hina Kosakamoto.

**Writing – original draft:** Taro Onuma, Toshitaka Yamauchi, Takumi Murakami, Fumiaki Obata.

**Writing – review & editing:** Taro Onuma, Toshitaka Yamauchi, Hina Kosakamoto, Hibiki Kadoguchi, Takayuki Kuraishi, Takumi Murakami, Masayuki Miura, Fumiaki Obata.

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
