## [Decision Letter · Decision Letter 0]

8 Dec 2022

Dear Dr Obata,

Thank you very much for submitting your Research Article entitled 'Recognition of commensal bacterial peptidoglycans defines Drosophila gut homeostasis and lifespan' to PLOS Genetics.

The manuscript was fully evaluated at the editorial level and by independent peer reviewers. The reviewers appreciated the attention to an important problem, but raised some substantial concerns about the current manuscript. Based on the reviews, we will not be able to accept this version of the manuscript, but we would be willing to review a much-revised version. We cannot, of course, promise publication at that time.

As you will see in their feedback the Reviewers found the approach using bacteria-conditioned diet potentially interesting. However, the Reviewers had many technical concerns regarding the approach (for example effect of antibiotics, quantity of bacteria), which should be addressed by additional control experiments. To specifically address the underlying reasons for the differential effects of bacterial strains, purification of the bacterial components (such as the peptidoglycans) should be conducted. This would serve as a strong proof of principle for the strategy of using bacteria-conditioned diet, making the study interesting to a broad audience.

If you decide to revise the manuscript for further consideration at PLOS Genetics, please aim to resubmit within the next 60 days, unless it will take extra time to address the concerns of the reviewers, in which case we would appreciate an expected resubmission date by email to plosgenetics@plos.org.

We are sorry that we cannot be more positive about your manuscript at this stage. Please do not hesitate to contact us if you have any concerns or questions.

Yours sincerely,

Ville Hietakangas

Academic Editor

PLOS Genetics

Gregory P. Copenhaver

Editor-in-Chief

PLOS Genetics

Reviewer's Responses to Questions

**Comments to the Authors:**

Reviewer #1: In this manuscript, Onuma and colleagues investigated how Acetobacter persici - a member of Drosophila melanogaster gut microbiota shortens the host lifespan. The authors used a bacteria-conditioned diet with antibiotics instead of alive bacteria to minimize contamination. Flies fed Acetobacter- but not Lactiplantibacillus- conditioned diet were short-lived, had increased intestinal stem cell proliferation and increased survival after intestinal infection with Pseudomonas entomophila. Transcriptomic analysis revealed that A. persici preferably induces antimicrobial peptides (AMPs),

while L. plantarum upregulates negative regulators of Imd pathway. Such differences in the induction of Imd target genes are likely due to differential activation of Imd pathway receptors, PGRP-LC and PGRP-LE by the two bacteria. This work suggests that differential ability of gut commensals to stimulate immune receptors results in distinct host response which determines the impact of gut bacteria on host lifespan. The authors performed an extensive comparative analysis to understand the differential impact of two gut commensals of the fruit fly lifespan. Additionally, they introduce a new technique - bacteria-conditioned diet to study microbial impact on host lifespan. Therefore, the manuscript is of interest to broad audience of Plos Genetics. However, certain issued still need to be resolved.

Comments:

1. The authors need to clearly state what control is in all their graphs.

2. For bacteria-conditioned medium it is claimed that only the culture medium was used. Was this medium treated with antibiotics similar to bacteria-inoculated food? This is an important point, since antibiotics might themselves affect the host lifespan.

3. Did the authors verify that their antibiotics cocktail actually kills all bacteria (Fig 1b)? This is essential to check (quantify viable bacteria after antibiotic treatment) since one possibility could be that Acetobacteriaceae are not killed by antibiotics and all the phenotypes that the authors observed are due to the presence or absence of alive bacteria.

4. Another import point to address is whether flies feed on such bacteria-conditioned diet especially after the addition of antibiotics. What if flies like Acetobacter-conditioned diet more then Lactobacillus-conditioned diet? They will eat more Acetobacter diet, therefore inducing stronger IMD response.

5. Fig 2E. Time should be days not hours.

6. The authors claim: "it is possible that

increased ISC proliferation in flies fed with the A. persici Ai-conditioned diet would be

beneficial for the acceleration of the damaged epithelia but detrimental for maintaining

healthy tissue during ageing." To provide further support for this claim ph3 counts should be performed after Pe infection.

Also, we do not know if Acetobacter promotes resistance or tolerance to Pe infection. If, as authors suggest, flies survive better because they have increased gut repair - this rather fulfills definition of tolerance. However, if it is due to stronger IMD activity and pathogen killing - it is resistance. Quantification of PE load will help to resolve this issue.

7. Better annotation of RNAseq table would be good. What does LvsM mean and the others? Did the authors found anything else in the RNAseq besides AMPs and PGRPs?

8. "To elucidate how A. persici Ai, but not L. plantarum Lsi, shortens the lifespan

and alters the host defence, we investigated the transcriptomic response of the gut to

each BacD." Better phrasing needed because this sounds like the authors will do RNAseq of aging animals, which is not what they have done.

9. "Interestingly, L. plantarum Lsi was also capable of inducing DptA from the posterior

midgut, although it did not reach statistical significance due to the large induction of

DptA by A. persici Ai (Fig 4C)." How does large induction of

DptA by A. persici affect statistical significance of DptA induction by L. plantarum?

10. "Therefore, we simply fed the flies heat-killed bacteria (Fig 6A)." Heat-killed bacteria still contain other things besides cell wall components. Therefore, attributing all the phenotypes with heat-killed bacteria to cell wall only or peptidoglycan is not correct. To make such claims, purified peptidoglycan has to be used.

11. Methods "Septic and oral infection of Pseudomonas entomophila"

Please specify the dose of bacteria used for pricking and oral infection. There is typo in P. entomophila.

Reviewer #2: Using Drosophila as a model, Onuma at al investigate the impact of bacteria-conditioned diet and heat-killed bacteria on several host physiological traits: lifespan, intestinal stem cell proliferation, infection, resistance to oxidative stress and activation of host defenses through IMD pathway. The authors identified a trade-off between lifespan and host defenses in presence of A. persici conditioned diets.

Although conceptually interesting, this work merely reports a set of initial interesting observations that deserve further analysis but at this stage the work raise several questions regarding the experimental designs.

The interesting observations that bacteria-conditioned diets influence fly lifespan differently given the bacteria used are currently embedded with a lot of experimental testing of already established facts about how Acetobacter spp and Lactobacillacae strains regulates AMPs and PGRPs at the transcriptional level in the adult fly gut.

One major concern refers to the use of ‘peptidoglycan’ to refer to ‘heat-killed bacteria’ throughout the manuscript including the title. It’s misleading and incorrect. To address this issue, I would suggest to extract peptidoglycan from the implicated strains, determine their composition (and structure) and test for the claimed phenotypes.

Secondly, the use of bacterial-conditioned diets is a rather artificial system. It’s possible to keep fly food tubes sterile during the time of an experiment, even for lifespan study (very clean and dedicated Fly incubators and ad hoc fly and tube handling procedures under a microbiology hood should be implemented). Nevertheless, concerning the experiments presented, it will be an added value to quantify the effect of the antibiotic cocktail on the bacterial CFUs and to produce a parallel between the bacterial-conditioned diets and metabolic active bacteria.

Finally, a deeper analytical study of the ”conditioned-diet” especially towards which microbial compounds and cell wall components are present is crucial to this work. A follow up on testing purified compounds is also necessary.

In general, legends deserve to be further completed, at least, those concerning the experimental protocols.

Reviewer #3: In this manuscript, Onuma et al. investigated the role of microbiome, with special interests in microbiome-derived factors, on fly aging. They optimized their assay by using bacteria-conditioned diet (BacD experiement). Using this method, they found that Acetobacter (i.e. Acetobacter-conditioned diet) shortened lifespan, which is consistent with their previous finding. In addition to lifespan, they also found that Acetobacter accelerates aging phenotypes such as ISC hyperproliferation and reduced climbing ability. Acetobacter increased host resistance to paraquat and oral pathogen infection, but reduced starvation stress. They further found that Acetobacter induces IMD-dependent AMP expression in the anterior midgut via PGRP-LC, whereas Lactobacillus induces IMD-dependent PGRP-SC1a gene in the posterior midgut via PGRP-LE. Heat-killed acetobacter also shortened lifespan, but did not affect stress phenotypes. Knockdown of IMD pathway revealed that Acetobacter-activated IMD pathway is responsible for ISC hyperproliferation and early death. Taken all together, they concluded that commensal Acetobacter-derived peptidoglycan plays and important role on intestinal cell homeostasis, gut immunity, and host lifespan.

General comments.

In general, it contains some interesting aspects of the peptidoglycan-activated immune sigaling pathway in the gut (e.g. regional specificities of PGRPs and caudal regulator). However, I am not sure that their experimental condition is physiologially relevant (although I agree that BacD experiment is more technically accessbile than conventional gnotobiotic experiment). As they tested all the experiment with BacD containing a cocktail of antibiotics, this extreme and artificial condition could not reflect the real physiological roles of commensal bacteria. Therefore, although I admit that all their data are solid in their experimental settings, I am not sure whether the conclusion of this work is still valid in a physiological condition.

Specific comments.

1) One of the problems in BacD experiment is that they use a single condition with relatively high number of bacteria. It may be helpful to see the result (e.g. survival) by using different number of bacteria (i.e. dose-dependent experiments).

2) It is often observed that antibiotics slightly affect the fitness of germ-free Drosophila. Do they test the effect of antibiotics?

3) It is unclear whether life-shortening effect is a general asepct of acetobacter. They should test different species of acetobacter (preferentially originated from other laboratories). Is there any difference between BacD experiement and gnotobiotic experiement?

4) Why and how does acetobacter affect host resistance against oxidative stress and starvation? What about the IMD knockdown animals (in ISC/EB as well as in EB/EC) under BacD experiment or under gnotobiotic experiement?

5) It is interesting to see the effect of Acetobacter on host overexpressing PGRP-SC in the posterior midgut.

6) It is unclear why Lactobacillus does not activate the IMD pathway in the anterior midgut. They should use peptidoglycan (partially purified from Lactobacillus) to see whether it could activate the IMD pathway in the anterior midgut.

7) Why do peptidoglycans of different bacteria use different receptors? Is it due to the different structure? What happens if they use partially-purified peptidoglycans from acetobacter and lactobacillus?

8) It is known that IMD pathway is not involved in ISC turnover (e.g. Buchon et al.). How does IMD pathway in ISC/EB affect ISC proliferation? Is enterocyte-specific IMD pathway involved in ISC turnover? In the same context, is ISC/EB-specific IMD pathway involved in lifespan?

9) In the text, they stated that “These metabolite alterations suggested that BacD would be suitable for understanding how bacteria and bacterial products influence their hosts and, thereby, possibly recapitulate the phenotypes seen in flies with a monoassociation” However, I think there are big differences between BacD and gnotobiotic experiment.

**Have all data underlying the figures and results presented in the manuscript been provided?**

Reviewer #1: Yes

Reviewer #2: Yes

Reviewer #3: Yes

PLOS authors have the option to publish the peer review history of their article (what does this mean?). If published, this will include your full peer review and any attached files.

Reviewer #1: No

Reviewer #2: **Yes: **François Leulier

Reviewer #3: No

---

## [Decision Letter · Decision Letter 1]

21 Mar 2023

Dear Dr Obata,

We are pleased to inform you that your manuscript entitled "Recognition of commensal bacterial peptidoglycans defines Drosophila gut homeostasis and lifespan" has been editorially accepted for publication in PLOS Genetics. Congratulations!

As you will see from the Reviewer feedback, one Reviewer expressed an opposing opinion regarding the importance of the further analysis of purified peptidoglycans. However, after carefully considering the Reviewer feedback as a whole, I have opted to accept the manuscript for publication.

Yours sincerely,

Ville Hietakangas

Academic Editor

PLOS Genetics

Gregory P. Copenhaver

Editor-in-Chief

PLOS Genetics

Comments from the reviewers (if applicable):

Reviewer's Responses to Questions

**Comments to the Authors:**

Reviewer #1: The authors have addressed my concerns.

Reviewer #2: The reviewer acknowledges the efforts made to purify and test the effect of PG on gene expression, however this effect on lifespan, although well argued by the authors, has not been addressed experimentally. I don’t agree that the analysis of the PG composition is out of scope, the authors claim that the impact on the phenotype is due to PG, thus what is different between them would be of interest, also for this publication.

IThe authors justify well their experimental set up concerning BacD diets. I’m satisfied with the arguments.

Reviewer #3: The authors have sufficiently addressed my concerns. I now recommend this manuscript for the publication.

**Have all data underlying the figures and results presented in the manuscript been provided?**

Reviewer #1: Yes

Reviewer #2: Yes

Reviewer #3: Yes

PLOS authors have the option to publish the peer review history of their article (what does this mean?). If published, this will include your full peer review and any attached files.

Reviewer #1: No

Reviewer #2: **Yes: **François Leulier

Reviewer #3: No

**Data Deposition**

http://datadryad.org/submit?journalID=pgenetics&manu=PGENETICS-D-22-01293R1

**Press Queries**

---

## [Editor Report · Acceptance letter]

31 Mar 2023

PGENETICS-D-22-01293R1 

Recognition of commensal bacterial peptidoglycans defines Drosophila gut homeostasis and lifespan 

Dear Dr Obata, 

We are pleased to inform you that your manuscript entitled "Recognition of commensal bacterial peptidoglycans defines Drosophila gut homeostasis and lifespan" has been formally accepted for publication in PLOS Genetics! Your manuscript is now with our production department and you will be notified of the publication date in due course.

With kind regards,

Anita Estes

PLOS Genetics

On behalf of:
